# Diffuse to Detect: Bi-Level Sample Rebalancing with Pseudo-Label Diffusion for Point-Supervised Infrared Small-Target Detection

**Zhu Liu** [1]   **Yuanhang Yao** [1]   **Ping Qian** [1]   **Zihang Chen** [1]   **Risheng Liu** [1]

## Abstract

Point supervision has become a scalable solution to address dense annotation for infrared small target detection, but its performance is limited by two coupled bottlenecks: unstable pseudo-label evolution in cluttered, low-contrast infrared imagery and severe sample-distribution imbalance. In this paper, we present a more adaptive and stable framework to address these issues. Leveraging the intrinsic consistency between thermal radiation patterns and heat diffusion, we propose a physics-induced annotation strategy that expands single-point labels into reliable pseudo-masks. To further enhance supervision and alleviate sample imbalance, we develop a bi-level dual-update framework that jointly optimizes detector weights, sample weights, and diffusion parameters. A meta-classifier dynamically predicts sample-wise loss weights, while a differentiable diffusion module refines pseudo-labels with detection feedback, enabling adaptive interaction between training and hyperparameter optimization. Extensive experiments across multiple datasets demonstrate five-fold annotation acceleration, superior detection accuracy, and comparable performance with 30% of the training data, validating the efficiency and practicality of our approach. Our code is available at https://github.com/yuanhang-yao/diffuse-to-detect.

## 1. Introduction

Infrared Small Target Detection (ISTD) is a long-standing and crucial task that aims to identify dim and tiny targets against cluttered backgrounds. As a component of infrared search-and-tracking systems, ISTD plays an indispensable role in monitoring (Zhang et al., 2022b; Li et al., 2023b), navigation (Wang et al., 2025), early warning (Xu et al., 2023), and maritime resource management (Zhang et al., 2022a). However, due to the long-range imaging mechanism, targets are often shapeless and textureless, comprising only about 0.15% of the total pixels, making them easily overlooked within complex and noisy backgrounds, which further complicates detection.

Recently, data-driven learning paradigms (Dai et al., 2021a;b) have emerged as the mainstream solution, employing fully supervised pipelines to map infrared observations to pixel-level dense annotations. To capture discriminative features (Jiang et al., 2025a), existing methods design architectures guided by prior knowledge (Zhang et al., 2025a; Huang et al., 2025), including densely nested structures (Li et al., 2023b), vision transformers (Chen et al., 2024), or fine-tuning of large foundational models (Zhang et al., 2024b;a). Despite their promising performance, these approaches typically demand precise annotations and large-scale datasets covering diverse, high-stakes scenarios. We argue that the development of learning-based ISTD faces two major obstacles. First, most approaches rely on pixel-level dense annotations (Li et al., 2023a), which are costly to obtain and prone to errors due to the inherent characteristics of infrared small targets (*e.g.*, low Signal-to-Noise Ratio (SNR) and indistinct boundaries). Second, the scarcity of high-quality ISTD datasets (Liu et al., 2025c) leads to severe long-tailed distributions, as illustrated in Fig. 1 (a), *e.g.*, only 200 and 800 training pairs in SIRST-v1 and IRSTD-1k. Such imbalance hinders sufficient training on rare but critical cases, thereby limiting generalization.

Existing studies (Chen et al., 2014a; Gao et al., 2013) have explored single-point supervision as an alternative to alleviate the annotation burden. These approaches can be broadly divided into two promising directions. The first category employs online label updates. For example, LESPS (Ying et al., 2023) evolves pseudo-labels through intermediate indicators but suffers from excessive evolution and instability, while PAL (Yu et al., 2025a) integrates active learning with a coarse-to-fine training scheme to improve stability, yet still relies heavily on manual parameter tuning that limits flexibility. In contrast, offline update strategies generate

---

[1]School of Software Technology, Dalian University of Technology, Dalian, China. Correspondence to: Risheng Liu <rsliu@dlut.edu.cn>.

*Proceedings of the 43$^{rd}$ International Conference on Machine Learning*, Seoul, South Korea. PMLR 306, 2026. Copyright 2026 by the author(s).

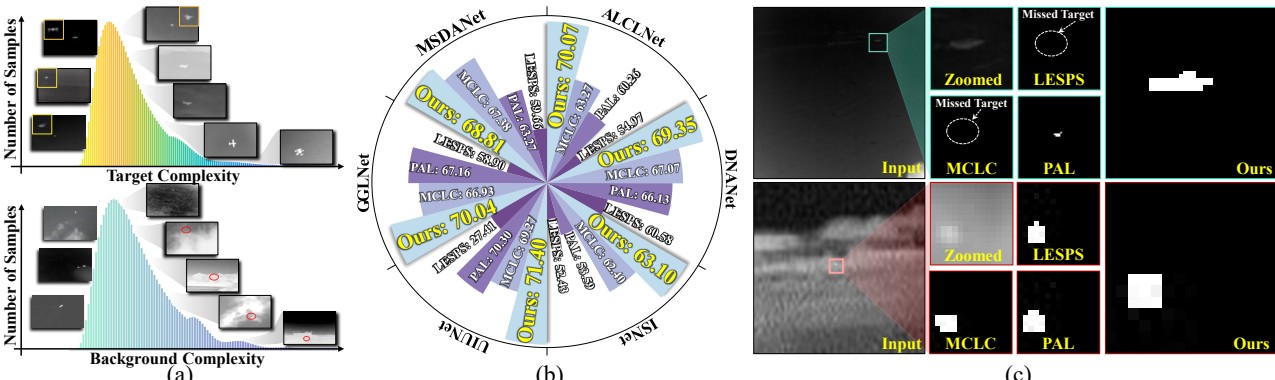

*Figure 1.* Motivation and efficiency overview. (a) Illustration of challenges including sample imbalance and complexity. (b) Quantitative results showing clear improvements using only 30% of the data. (c) Qualitative results validating our robustness in complex scenes.

pseudo-labels through dedicated algorithms (Li et al., 2023a; Kou et al., 2024), such as MCLC (Li et al., 2023a), which constructs labels from repeated noisy perturbations based on Monte Carlo linear clustering. However, these methods remain vulnerable to false positives, where background noise is mistakenly identified as targets. These limitations highlight the need for a more stable and precise automatic pseudo-label generation strategy.

Furthermore, several attempts have been made to alleviate the extreme sample imbalance in ISTD. Static loss reweighting schemes, such as focal loss (Xu et al., 2023), truncated squeeze loss (Zhang et al., 2025b) and adversarial learning (Wang et al., 2019) remain fixed throughout training. Curriculum-style methods, exemplified by progressive active learning, gradually introduce samples from easy to hard in a coarse-to-fine manner (Yu et al., 2025a), yet still rely on manually crafted schedules and heuristic criteria. Despite their merits, these approaches lack adaptability to the evolving training dynamics.

To address the aforementioned challenges, we propose a more adaptive and stable ISTD framework under single-point supervision. Leveraging the natural consistency between radiation patterns of infrared targets and physical heat diffusion, we first introduce an offline physics-induced annotation strategy that treats each spot as a thermal source, with diffusion governing the spatial spread of energy. To further refine supervision and reduce errors in complex cases, we develop a bi-level dual-update framework from the perspective of hyperparameter optimization. In this framework, the inner-level optimization updates the detector weights by minimizing the training loss, while the outer-level optimization adaptively adjusts two types of hyperparameters: sample weights and diffusion parameters. Specifically, a meta-classifier predicts sample-wise loss weights conditioned on training dynamics, enabling adaptive rebalancing of sample contributions. In parallel, the diffusion process is param-

eterized as a differentiable module, whose parameters are updated by the outer-level objective to ensure pseudo-labels evolve consistently with validation feedback. Furthermore, we design a dynamic aggregated solution to coordinate the interaction between model training and hyperparameter optimization. Extensive experiments demonstrate that our approach accelerates annotation by five times, achieves superior performance across multiple benchmarks, and attains comparable results using only 30% of the training data. The core contributions can be summarized as follows:

- We propose a bi-level dual-update optimization framework that jointly optimizes detector weights, sample weights, and annotation quality, achieving greater stability and adaptability than heuristic adjustments.

- By integrating a meta-classifier for sample-wise loss weighting with a differentiable diffusion module, our framework simultaneously rebalances samples and refines pseudo-labels under detection feedback.

- This work also introduces a physics-guided methodology for pseudo-mask generation, leveraging the intrinsic consistency between radiation patterns and heat diffusion to achieve efficient single-point supervision.

- Extensive experiments across multiple ISTD benchmarks validate the efficiency (annotation acceleration), effectiveness (superior detection accuracy), and practicality (fewer training samples) of our approach.

## 2. Related Work

**SIRST detection** has been widely studied for decades. Early research focused on model-driven traditional methods, including local contrast measurement methods (Chen et al., 2014a;b), filtering-based methods (Gao et al., 2013), and low-rank methods (He et al., 2015; Zhu et al., 2020).

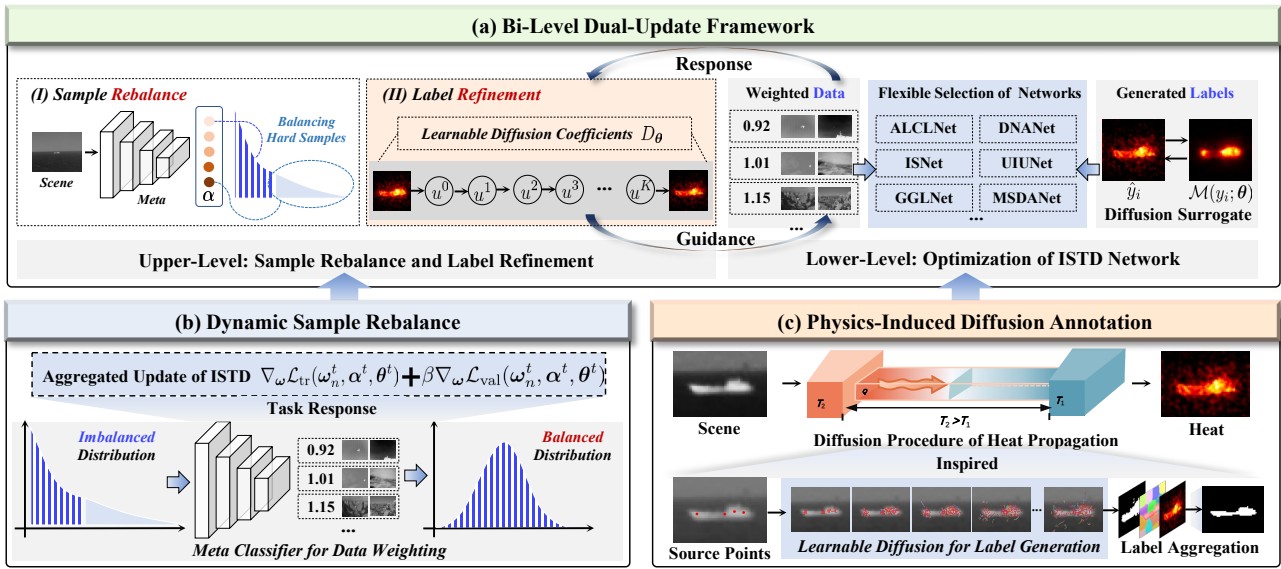

*Figure 2.* Overview of the proposed framework. (a) Bi-level dual-update framework performs joint sample rebalancing and label refinement. (b) Dynamic sample rebalancing is designed to weight training data. (c) Physics-induced diffusion annotation generates reliable pseudo-masks from single-point supervision in a learnable manner.

However, these methods require careful model design and fine-tuning of hyperparameters, resulting in poor adaptability and difficulty in dealing with complex and ever-changing real-world scenarios (Liu et al., 2023b). The subsequent emergence of deep learning methods (Zhao et al., 2022; Chen et al., 2023), especially fully supervised methods (Lu et al., 2024) based on customized architectures, has enabled models to learn the nonlinear mapping between input images (Zhao et al., 2022) and labels in a data-driven manner (Liu et al., 2024a). Various mechanisms including context aggregation (Lu et al., 2025), edge-guidance (Li et al., 2024b), dense feature representation (Li et al., 2023b), and spatial-frequency interaction, have been proposed (Liu et al., 2025b). Recently, several works have introduced granularity-aware modules and textual prompts into visual foundation models (*e.g.,* SAM (Kirillov et al., 2023)). Unfortunately, these methods are limited by the scarcity of large-scale datasets, which limits generalization.

**Point-based supervision** has been proposed for diverse visual perception tasks, such as object detection (Ying et al., 2023), localization (Li et al., 2023b), and segmentation. For instance, in oriented object detection, Wholly-WOOD (Yu et al., 2025c) studies unified weakly supervised oriented object detection with Point/HBox/RBox annotations, PointOBB (Luo et al., 2024) focuses on recovering object scale and angle from point supervision for oriented detection, and Point2RBox-v2 (Yu et al., 2025b) improves point-supervised oriented detection by modeling spatial layout among instances. At present, point-level segmentation methods require general targets with rich colors (Gao

et al., 2013), fine textures, and multiple annotated points, which pose challenges for the ISTD task. Ying *et al.* (Ying et al., 2023) proposed the LESPS framework that implements single-point supervised SIRST detection. However, the LESPS framework has issues such as instability and excessive label evolution. Li *et al.* (Li et al., 2023a) transformed fully supervised SIRST detection into a weakly supervised network with single-point supervision using the Monte Carlo clustering method, achieving remarkable performance while reducing annotation burden. Yu *et al.* (Yu et al., 2025a) proposed the PAL framework, which enables the initial selection of simple samples and the generation of corresponding pseudo-labels, effectively alleviating the problem of excessive label evolution. However, it lacks adaptability to the constantly evolving training dynamics and still requires adjustments. Thus, we aim to construct an efficient and adaptable framework.

## 3. The Proposed Method

### 3.1. Bi-Level Dual-Update Framework

In point-supervised ISTD, learning is fundamentally constrained by two coupled bottlenecks, including unstable pseudo-label evolution and severe sample-distribution imbalance. Importantly, the foundational issues amplify each other: imbalance makes the model more susceptible to overfitting to spurious background patterns in pseudo-labels, while noisy pseudo-labels further corrupt the estimation of sample importance, destabilizing the training dynamics. To overcome these issues, we introduce a bi-level dual-update

framework that jointly optimizes sample balance and supervision evolution based on the combination of offline and online updates, as shown in Fig. 2 (a). Hyperparameter optimization provides an effective tool for adaptively refining supervision according to model feedback, while the hierarchical separation enhances optimization stability and prevents error accumulation from noisy labels. Thus, we define the bi-level formulation (Liu et al., 2024c;b) as:

$$
\begin{aligned}
\min_{\boldsymbol{\alpha},\boldsymbol{\theta}} \quad & \mathcal{L}_{\text{val}}\big(\boldsymbol{\omega}^*(\boldsymbol{\alpha},\boldsymbol{\theta}),\boldsymbol{\alpha},\boldsymbol{\theta}\big) \\
\text{s.t.} \quad & \boldsymbol{\omega}^*(\boldsymbol{\alpha},\boldsymbol{\theta}) = \arg\min_{\boldsymbol{\omega}} \sum_i \alpha_i \, \mathcal{L}_{\text{tr}}\big(\mathcal{N}(x_i;\boldsymbol{\omega}),\hat{y}_i\big),
\end{aligned} \tag{1}
$$

where $\mathcal{N}$ denotes the detector with parameters $\boldsymbol{\omega}$, and $x_i$ and $\hat{y}_i(\boldsymbol{\theta})$ denote the input infrared image and its physics-induced pseudo-label, respectively. The variables $\boldsymbol{\alpha}$ and $\boldsymbol{\theta}$ correspond to two parallel hyperparameter branches in our framework. Specifically, the inner-level problem updates the detector parameters $\boldsymbol{\omega}$ by minimizing the weighted training loss, where $\boldsymbol{\theta}$ controls the diffusion-based annotation process and $\boldsymbol{\alpha}$ denotes the sample-wise loss weights. The outer-level objective then refines both $\boldsymbol{\alpha}$ and $\boldsymbol{\theta}$ by minimizing the validation loss $\mathcal{L}_{\text{val}}$, which reflects how well the detector generalizes under the current supervision (Fig. 2 (b)). Note that, instead of leveraging real labels, we leverage the physics-induced pseudo-labels for the validation set. Compared with existing single-point supervised methods, the proposed bi-level dual-update framework establishes a dynamic closed-loop between model training and supervision generation, enabling continuous feedback to correct noisy labels and prevent overfitting.

## 3.2. Physics-Induced Diffusion Annotation

Infrared small targets are characterized by localized thermal emissions, small spatial size, and low signal-to-noise ratio (SNR). Most existing label-generation approaches often neglect the underlying physics of thermal propagation, failing to capture the spatial energy distribution. Physically, an infrared target acts as a local heat source, radiating energy that diffuses outward (Liu et al., 2016; Metzger et al., 2023). This diffusion is intrinsically governed by the local thermal conductivity of the surrounding medium (*e.g.,* texture and material), causing the energy to attenuate (Borgnakke & Sonntag, 2020). This motivates our formulation of pseudo-label generation as a learnable thermal diffusion process. We model the pseudo-label $\hat{y}_i$ for the $i$-th target, identified by its single-point annotation $p_i$, as the quasi-steady thermal potential field $u(a,b)$ resulting from energy diffusion from the source $p_i$. The spatio-temporal evolution of this field, $u(a,b,t)$, is governed by a learnable anisotropic diffusion equation (Bao et al., 2023; Wang et al., 2023):

$$
\frac{\partial u(a,b,t)}{\partial t} = \nabla \cdot \big(D_{\boldsymbol{\theta}}(a,b)\nabla u(a,b,t)\big), \tag{2}
$$

with the initial condition $u(a,b,0) = \delta(a-p_i,b-p_i)$. Here, $D_{\boldsymbol{\theta}}(a,b)$ is the spatially varying diffusion tensor with learnable parameters $\boldsymbol{\theta}$. It governs the local thermal conductivity and modulates the diffusion rate according to regional texture and contrast: propagation accelerates in homogeneous areas (to fill missing regions) while decelerating near edges and high-gradient structures (to prevent boundary smearing). For computational tractability, we discretize Eq. (2) into an iterative form:

$$
u^{(k+1)} = (I - \tau L_{D_{\boldsymbol{\theta}}})u^{(k)}, \quad u^{(0)} = \delta_{p_i}, \tag{3}
$$

where $u^{(k)}$ is the thermal potential after the $k$-th iteration, $\tau$ is the diffusion step size, and $L_{D_{\boldsymbol{\theta}}}$ is an image-adaptive Laplacian operator constructed from $D_{\boldsymbol{\theta}}(a,b)$. After $K$ iterations, $u^{(K)}$ represents the quasi-steady thermal field.

To enhance structural integrity and robustness, we aggregate this physics-based field with a data-driven spatial prior. Specifically, we introduce a superpixel segment $C_{p_i}(a,b)$, which is the binary mask of the superpixel region containing the annotation $p_i$. This prior provides a compact, data-driven boundary based on local image statistics. We then define the final pseudo-label $\hat{y}_i$ as a learnable aggregation of these two information sources:

$$
\hat{y}_i(\boldsymbol{\theta}) = \rho \cdot u^{(K)}(a,b) + (1-\rho) \cdot C_{p_i}(a,b), \tag{4}
$$

where $\rho \in [0,1]$ is a learnable balance parameter. This fusion adaptively controls the trade-off between the physics-informed diffusion $u^{(K)}$ and the compact structural prior $C_{p_i}$. The aggregated field $\hat{y}_i(\boldsymbol{\theta})$ is subsequently normalized to $[0,1]$. The final pseudo-label $\hat{y}_i(\boldsymbol{\theta})$ thus preserves both physical continuity and structural precision, serving as a differentiable supervision signal for the bi-level optimization framework, as shown in Fig. 2 (c).

## 3.3. Dynamic Aggregated Solution

In practice, solving the hyperparameter optimization in Eq. (1) is computationally demanding, as it typically requires high-order unrolled gradients (Liu et al., 2025a) or implicit differentiation (Liu et al., 2022). Although alternating updates can reduce computational burden, they often lead to unstable convergence due to the mismatched update frequencies between the inner and outer levels. To overcome these limitations, we introduce a first-order dynamic aggregated solution (Liu et al., 2023a) that enables efficient and coherent interaction between model parameters and hyperparameters within a unified optimization.

Inspired by recent work on single-loop bi-level optimization (Jiang et al., 2025b), we dynamically combine these two objectives to update the model parameters in the inner loop to obtain $\boldsymbol{\omega}^{t+1}$: $\boldsymbol{\omega}_{n+1}^t = \boldsymbol{\omega}_n^t - \Big(\nabla_{\boldsymbol{\omega}}\mathcal{L}_{\text{tr}}(\boldsymbol{\omega}_n^t,\boldsymbol{\alpha}^t,\boldsymbol{\theta}^t) + \beta\,\nabla_{\boldsymbol{\omega}}\mathcal{L}_{\text{val}}(\boldsymbol{\omega}_n^t,\boldsymbol{\alpha}^t,\boldsymbol{\theta}^t)\Big)$, where $\beta$ controls how strongly the

*Table 1.* Performance of ISTD under different training strategies. All models are trained on SIRST3 and evaluated on SIRST3 and three individual datasets (SIRST-v1, NUDT-SIRST, IRSTD-1k). We report four metrics: IoU (%), nIoU (%), $P_d$ (%), and $F_a$ ($10^{-6}$).

| Scheme | Description | SIRST3 | | | | SIRST-v1 | | | | NUDT-SIRST | | | | IRSTD-1k | | | |
|---|---|---|---|---|---|---|---|---|---|---|---|---|---|---|---|---|---|
| | | IoU↑ | nIoU↑ | $P_d$↑ | $F_a$↓ | IoU↑ | nIoU↑ | $P_d$↑ | $F_a$↓ | IoU↑ | nIoU↑ | $P_d$↑ | $F_a$↓ | IoU↑ | nIoU↑ | $P_d$↑ | $F_a$↓ |
| MLCLNet (Yu et al., 2022a) | Full | 81.04 | 83.56 | 95.95 | 14.07 | 69.01 | 75.37 | 92.78 | 62.56 | 91.25 | 91.54 | 97.67 | 10.09 | 64.29 | 64.22 | 92.93 | 7.91 |
| | LESPS | 35.02 | 44.81 | 69.50 | 28.22 | 44.93 | 52.68 | 77.95 | 33.79 | 38.71 | 46.31 | 70.16 | 27.16 | 26.75 | 41.49 | 71.04 | 22.75 |
| | PAL | 64.17 | 68.48 | 94.85 | 18.26 | 63.48 | 68.41 | 92.02 | 22.43 | 66.68 | 72.12 | 96.88 | 22.09 | 63.11 | 56.34 | 89.24 | 17.63 |
| | MCLC | 65.16 | 67.95 | 96.22 | 30.53 | 64.83 | 68.17 | 89.92 | 40.41 | 68.51 | 71.23 | 94.07 | 31.80 | 55.44 | 56.30 | 87.58 | 14.90 |
| | Ours | 69.95 | 72.61 | 94.57 | 11.12 | 65.70 | 69.39 | 92.59 | 7.90 | 78.68 | 85.70 | 97.67 | 12.79 | 60.19 | 56.94 | 90.12 | 11.20 |
| ALCLNet (Yu et al., 2022b) | Full | 82.12 | 82.57 | 96.84 | 13.36 | 75.65 | 75.91 | 96.20 | 24.90 | 88.78 | 89.26 | 97.67 | 10.23 | 64.31 | 64.31 | 91.25 | 12.13 |
| | LESPS | 30.59 | 43.91 | 70.30 | 34.36 | 42.61 | 53.35 | 76.81 | 37.65 | 31.33 | 43.47 | 70.26 | 37.87 | 25.61 | 42.18 | 71.04 | 27.27 |
| | PAL | 60.26 | 64.67 | 91.89 | 35.34 | 64.19 | 68.99 | 92.78 | 30.80 | 63.67 | 65.98 | 93.12 | 40.84 | 42.86 | 54.08 | 86.63 | 26.14 |
| | MCLC | 63.27 | 65.70 | 91.89 | 47.19 | 68.09 | 72.41 | 93.54 | 51.86 | 63.01 | 65.58 | 91.22 | 49.25 | 58.15 | 55.94 | 88.26 | 34.47 |
| | Ours | 73.28 | 73.30 | 95.44 | 16.35 | 74.34 | 72.92 | 98.15 | 13.82 | 79.94 | 85.50 | 97.67 | 10.04 | 63.96 | 57.66 | 92.18 | 24.29 |
| DNANet (Li et al., 2023b) | Full | 85.49 | 86.22 | 96.88 | 8.14 | 77.95 | 80.35 | 96.58 | 22.23 | 93.90 | 94.06 | 98.84 | 3.06 | 65.21 | 66.64 | 91.25 | 10.21 |
| | LESPS | 36.98 | 47.59 | 86.84 | 28.97 | 44.39 | 52.49 | 84.79 | 30.10 | 37.27 | 46.40 | 90.58 | 26.04 | 31.97 | 47.92 | 76.43 | 27.80 |
| | PAL | 66.13 | 70.37 | 94.76 | 21.38 | 67.28 | 70.91 | 95.06 | 22.09 | 71.23 | 73.53 | 96.88 | 22.24 | 48.89 | 57.38 | 89.27 | 23.85 |
| | MCLC | 67.07 | 69.29 | 94.22 | 33.03 | 68.08 | 69.08 | 95.96 | 37.32 | 70.22 | 73.03 | 94.71 | 35.46 | 56.92 | 56.66 | 90.57 | 23.55 |
| | Ours | 71.08 | 74.65 | 95.24 | 15.16 | 68.27 | 73.49 | 96.30 | 14.90 | 79.99 | 87.01 | 97.67 | 13.65 | 61.38 | 58.07 | 90.82 | 17.00 |
| ISNet (Zhang et al., 2022b) | Full | 74.38 | 76.34 | 93.29 | 28.10 | 70.47 | 72.89 | 92.78 | 41.23 | 80.19 | 81.73 | 94.18 | 22.41 | 61.46 | 60.52 | 89.23 | 32.13 |
| | LESPS | 32.22 | 44.20 | 77.94 | 36.98 | 40.71 | 51.00 | 82.13 | 38.59 | 35.46 | 45.63 | 79.15 | 35.02 | 24.02 | 38.40 | 71.72 | 36.74 |
| | PAL | 53.59 | 58.42 | 86.11 | 35.57 | 53.01 | 60.53 | 88.21 | 32.38 | 57.69 | 62.15 | 88.68 | 34.75 | 31.92 | 45.23 | 77.78 | 39.89 |
| | MCLC | 62.40 | 64.07 | 90.42 | 44.53 | 65.12 | 68.04 | 94.68 | 27.71 | 65.13 | 66.88 | 92.76 | 55.59 | 52.65 | 52.35 | 85.58 | 32.57 |
| | Ours | 65.83 | 69.39 | 91.02 | 19.29 | 66.82 | 71.14 | 96.30 | 23.21 | 72.60 | 79.91 | 93.02 | 9.24 | 54.81 | 53.53 | 86.39 | 30.59 |
| UIUNet (Wu et al., 2023) | Full | 85.21 | 85.73 | 97.01 | 11.86 | 76.74 | 78.53 | 95.82 | 21.82 | 93.57 | 93.75 | 98.62 | 4.69 | 68.22 | 66.57 | 93.27 | 20.95 |
| | LESPS | 39.38 | 50.20 | 75.81 | 33.43 | 53.06 | 60.58 | 82.89 | 29.76 | 40.17 | 49.04 | 75.24 | 30.32 | 32.37 | 47.56 | 76.09 | 36.25 |
| | PAL | 70.30 | 71.95 | 94.48 | 25.29 | 72.15 | 74.15 | 95.44 | 34.64 | 71.95 | 73.71 | 97.31 | 20.87 | 65.82 | 62.21 | 91.95 | 31.35 |
| | MCLC | 69.27 | 71.07 | 94.81 | 37.60 | 72.45 | 73.76 | 96.20 | 42.26 | 69.81 | 73.35 | 96.08 | 36.97 | 65.15 | 63.79 | 91.93 | 36.31 |
| | Ours | 73.65 | 75.60 | 96.31 | 15.40 | 73.00 | 74.22 | 100.00 | 11.49 | 74.59 | 84.22 | 97.67 | 15.95 | 65.86 | 64.75 | 92.16 | 16.93 |
| GGLNet (Zhao et al., 2023) | Full | 84.08 | 85.38 | 97.61 | 8.58 | 78.82 | 79.45 | 97.34 | 25.93 | 92.33 | 92.86 | 99.26 | 3.65 | 64.52 | 65.54 | 91.58 | 8.27 |
| | LESPS | 40.89 | 50.95 | 74.42 | 33.34 | 50.07 | 59.07 | 79.85 | 33.94 | 43.61 | 51.04 | 76.19 | 37.03 | 32.26 | 46.75 | 71.04 | 28.67 |
| | PAL | 67.16 | 70.42 | 94.95 | 27.19 | 68.81 | 71.78 | 96.20 | 27.51 | 71.07 | 72.71 | 97.46 | 26.93 | 53.08 | 59.54 | 89.23 | 27.19 |
| | MCLC | 66.93 | 68.64 | 93.15 | 26.02 | 70.20 | 71.94 | 97.72 | 21.40 | 68.24 | 70.26 | 93.23 | 26.34 | 56.82 | 58.87 | 88.92 | 29.98 |
| | Ours | 71.73 | 75.35 | 95.04 | 16.23 | 72.55 | 73.37 | 98.15 | 11.13 | 82.20 | 88.33 | 98.26 | 14.63 | 58.24 | 59.91 | 89.80 | 20.48 |
| MSDANet (Zhao et al., 2025) | Full | 86.16 | 86.13 | 97.01 | 11.04 | 76.40 | 78.03 | 94.30 | 25.79 | 94.01 | 94.18 | 98.73 | 1.47 | 71.64 | 68.09 | 93.94 | 28.26 |
| | LESPS | 39.03 | 49.84 | 82.86 | 25.58 | 50.27 | 58.15 | 85.17 | 22.43 | 39.60 | 48.67 | 84.23 | 24.82 | 33.11 | 47.70 | 78.11 | 29.42 |
| | PAL | 63.27 | 68.68 | 94.15 | 23.29 | 67.76 | 71.34 | 94.68 | 16.81 | 67.48 | 70.61 | 96.83 | 23.76 | 47.80 | 58.02 | 89.23 | 27.54 |
| | MCLC | 67.38 | 69.88 | 93.82 | 37.35 | 70.61 | 72.31 | 96.96 | 14.13 | 68.94 | 71.67 | 94.81 | 47.68 | 58.84 | 58.71 | 89.93 | 27.33 |
| | Ours | 71.41 | 73.16 | 94.37 | 12.71 | 71.99 | 72.51 | 97.22 | 7.54 | 79.25 | 85.38 | 97.09 | 5.85 | 60.61 | 59.38 | 90.82 | 23.76 |

outer-level objective influences the inner-level parameter updates, achieving a balance between leveraging validation feedback and maintaining training stability. This dynamic aggregation allows the model parameters to evolve under both task-specific supervision and meta-level feedback. Subsequently, the hyperparameters $\boldsymbol{\alpha}$ are updated in a one-step gradient manner based on the latest model parameters $\boldsymbol{\omega}^{t+1}$: $\boldsymbol{\alpha}^{t+1} = \boldsymbol{\alpha}^t - \nabla_{\boldsymbol{\alpha}}\mathcal{L}_{\text{val}}(\boldsymbol{\omega}^{t+1}, \boldsymbol{\alpha}^t, \boldsymbol{\theta}^t)$. This update strategy ensures that the learned loss weights are adaptively aligned with the outer-level objectives. Finally, we refine the diffusion parameters $\boldsymbol{\theta}$ by aligning the differentiable diffusion surrogate $\mathcal{M}$ with the physics-induced pseudo-labels produced by the offline annotator. Given the detector prediction $y_i = \mathcal{N}(x_i; \boldsymbol{\omega})$, the surrogate diffusion module outputs a soft mask $\mathcal{M}(y_i; \boldsymbol{\theta})$, while $\hat{y}_i$ denotes the corresponding pseudo-label generated by the basic physics-induced diffusion process. We update $\boldsymbol{\theta}$ with a single gradient step: $\boldsymbol{\theta}^{t+1} = \boldsymbol{\theta}^t - \nabla_{\boldsymbol{\theta}}\Big(\mathcal{L}_{\text{val}}\big(\mathcal{M}(y_i; \boldsymbol{\theta}), \hat{y}_i\big) + \mathcal{L}_{\text{val}}\big(\mathcal{M}(y_i; \boldsymbol{\theta}), y_i\big)\Big)$. Here, the first term encourages $\mathcal{M}$ to reproduce the physics-induced pseudo-labels, while the second term regularizes the update by keeping $\mathcal{M}(y_i; \boldsymbol{\theta})$ consistent with the current response, mitigating overfitting

to potentially noisy supervision. This design ensures that the outer objective provides a stable signal for generalization-oriented weighting and hyperparameter calibration, while preventing the validation loss from collapsing into self-consistency with the current predictions. The overall procedure of this framework is summarized in Alg. 1.

## 4. Experiments

### 4.1. Implementation Configurations

**Datasets.** We evaluate on four representative datasets: SIRST3 (Ying et al., 2023), SIRST-v1 (Dai et al., 2021a), NUDT-SIRST (Li et al., 2023b), and IRSTD-1k (Zhang et al., 2022b), which contain 2,755, 427, 1,327, and 1,001 samples, respectively. Since there is no common practice of using a validation set on these benchmarks, we adopt a 6:2:2 partition for each of SIRST-v1, NUDT-SIRST, and IRSTD-1k, following the official splitting protocols (Dai et al., 2021a; Li et al., 2023b; Zhang et al., 2022b). For SIRST3, we merge the three subsets and perform a 6:2:2 split for train/val/test to evaluate robustness on diverse scenes.

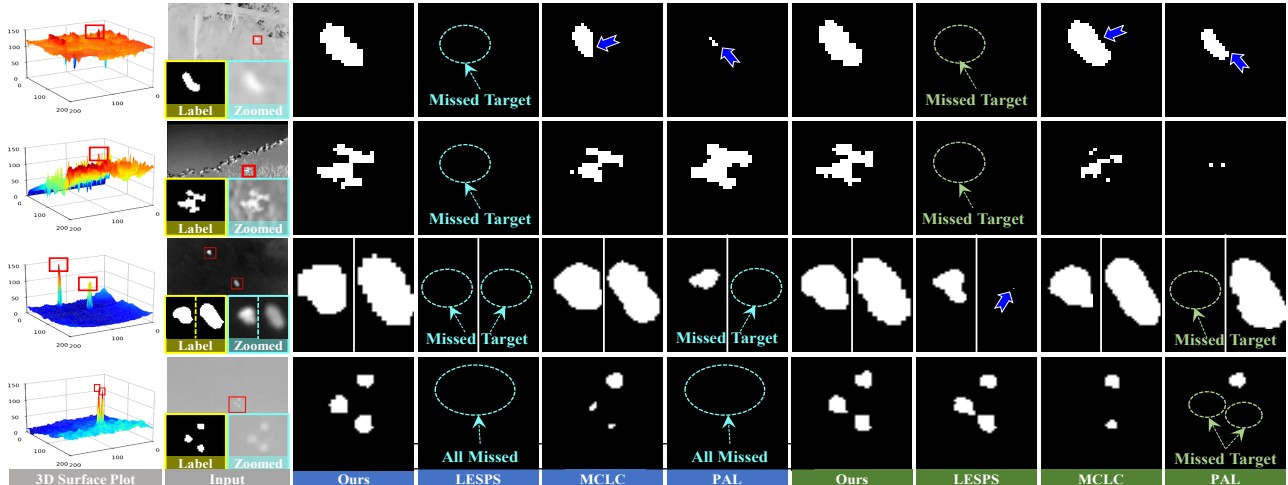

*Figure 3.* Qualitative comparison on the SIRST3 dataset. Columns from left to right: 3D surfaces of inputs, input images with labels and local magnification, Ours, LESPS, MCLC, and PAL predictions (the blue part on ALCLNet, the green part on DNANet).

*Table 2.* Performance of ISTD under different training strategies. Models are trained on three individual datasets (SIRST-v1, NUDT-SIRST, IRSTD-1k). We report four metrics: IoU (%), nIoU (%), $P_d$ (%), and $F_a$ ($10^{-6}$).

| Methods | Strategy | SIRST-v1 | | | | NUDT-SIRST | | | | IRSTD-1k | | | |
|---|---|---|---|---|---|---|---|---|---|---|---|---|---|
| | | IoU↑ | nIoU↑ | $P_d$↑ | $F_a$↓ | IoU↑ | nIoU↑ | $P_d$↑ | $F_a$↓ | IoU↑ | nIoU↑ | $P_d$↑ | $F_a$↓ |
| ALCLNet (Yu et al., 2022b) | Full | 72.57 | 72.66 | 94.68 | 30.39 | 89.79 | 89.95 | 98.84 | 6.69 | 65.23 | 64.25 | 89.90 | 9.19 |
| | LESPS | 39.85 | 43.16 | 73.76 | 35.37 | 36.76 | 44.90 | 73.02 | 39.02 | 33.73 | 47.75 | 80.81 | 36.21 |
| | PAL | 60.49 | 62.48 | 86.69 | 43.63 | 67.40 | 68.89 | 96.61 | 23.00 | 57.48 | 55.40 | 87.54 | 27.86 |
| | MCLC | 71.80 | 71.71 | 95.06 | 33.07 | 68.89 | 70.84 | 95.45 | 39.94 | 63.11 | 59.66 | 91.58 | 29.80 |
| | Ours | 75.62 | 71.93 | 98.15 | 2.51 | 79.88 | 85.50 | 96.80 | 17.27 | 64.72 | 60.35 | 92.52 | 18.07 |
| DNANet (Li et al., 2023b) | Full | 77.55 | 77.76 | 94.30 | 11.66 | 95.35 | 95.43 | 99.26 | 2.00 | 67.74 | 63.66 | 88.55 | 17.56 |
| | LESPS | 54.39 | 61.13 | 89.35 | 30.18 | 36.98 | 46.46 | 66.14 | 31.97 | 30.59 | 47.47 | 72.39 | 29.60 |
| | PAL | 64.83 | 66.94 | 92.78 | 30.87 | 70.70 | 71.34 | 97.78 | 24.57 | 62.29 | 58.46 | 87.88 | 20.65 |
| | MCLC | 69.61 | 72.21 | 96.96 | 24.08 | 70.86 | 72.82 | 96.19 | 25.00 | 61.79 | 64.32 | 90.91 | 32.51 |
| | Ours | 75.49 | 72.94 | 99.07 | 8.08 | 79.32 | 87.20 | 98.26 | 10.21 | 63.64 | 64.67 | 92.59 | 18.67 |
| UIUNet (Wu et al., 2023) | Full | 80.03 | 78.99 | 94.68 | 9.12 | 95.38 | 95.22 | 99.05 | 1.38 | 70.36 | 64.09 | 93.60 | 18.24 |
| | LESPS | 46.82 | 55.18 | 78.71 | 26.40 | 39.63 | 46.56 | 68.04 | 20.04 | 39.90 | 49.67 | 80.47 | 20.70 |
| | PAL | 63.98 | 62.39 | 92.02 | 28.95 | 70.81 | 72.43 | 98.20 | 13.42 | 57.50 | 56.98 | 89.90 | 11.86 |
| | MCLC | 71.01 | 68.87 | 90.11 | 22.91 | 70.20 | 72.22 | 94.71 | 21.46 | 66.19 | 62.31 | 88.89 | 39.30 |
| | Ours | 71.85 | 69.55 | 99.07 | 11.31 | 79.84 | 85.19 | 98.55 | 6.77 | 68.19 | 62.86 | 90.14 | 6.07 |
| MSDANet (Zhao et al., 2025) | Full | 75.82 | 76.49 | 95.06 | 38.49 | 95.65 | 95.48 | 99.47 | 1.38 | 68.08 | 64.29 | 91.92 | 24.05 |
| | LESPS | 48.69 | 55.86 | 82.89 | 32.12 | 36.34 | 45.45 | 73.76 | 19.40 | 38.73 | 49.79 | 81.48 | 33.55 |
| | PAL | 63.27 | 66.92 | 93.92 | 29.64 | 59.46 | 63.76 | 93.54 | 16.13 | 59.52 | 58.16 | 89.90 | 26.94 |
| | MCLC | 71.23 | 72.35 | 92.02 | 31.83 | 69.67 | 71.61 | 94.92 | 26.52 | 62.83 | 61.77 | 87.54 | 32.32 |
| | Ours | 76.33 | 73.79 | 100.00 | 3.77 | 79.08 | 86.05 | 98.84 | 13.19 | 63.31 | 62.94 | 90.12 | 25.98 |

**Experimental settings.** We trained for 400 epochs using the AdamW optimizer with a batch size of 16 and a learning rate of $1 \times 10^{-3}$. Outer-level update activates at epoch 80, updating the sample-weighting and pseudo-label branches in parallel. The pseudo-label branch updates every 20 epochs via a separate Adam optimizer with learning rate $1 \times 10^{-2}$. Inputs were normalized and randomly cropped to $256 \times 256$. Soft IoU loss is used to define $\mathcal{L}_{\text{val}}$ and $\mathcal{L}_{\text{tr}}$. Both the training and validation sets are supervised only by single-point annotations, and all image priors used in our method are computed solely from the input infrared images. We report standard SIRST metrics (IoU, nIoU, $P_d$, $F_a$) to assess segmentation fidelity, missed targets, and false positives. Network details of meta-classifier and learnable diffusion are provided in appendix.

## 4.2. Experimental Results

**Evaluation on the SIRST3 dataset.** Table 1 presents a comprehensive performance comparison of our proposed training strategy against leading single-point supervision baselines (LESPS, PAL, MCLC) and a fully supervised (Full) upper bound. The evaluation spans seven different detector architectures on the comprehensive SIRST3 benchmark. The results definitively demonstrate that our approach consistently and significantly outperforms all competing single-point supervision methods across nearly every metric and detector. This trend of superiority is consistent across all benchmarks; on SIRST3, our strategy achieves the highest IoU and nIoU for every evaluated detector among all non-full supervision methods.

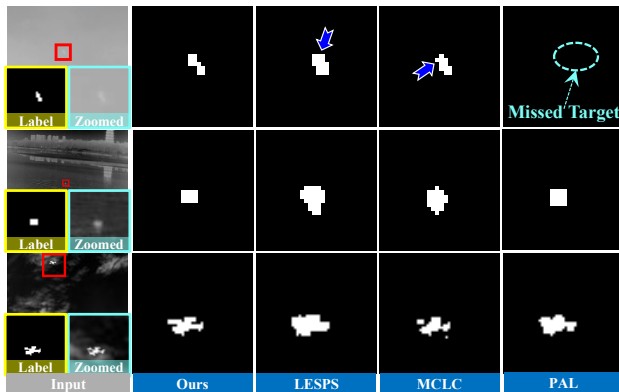

*Figure 4.* Qualitative comparison on three representative scenes. These results are obtained by ALCLNet.

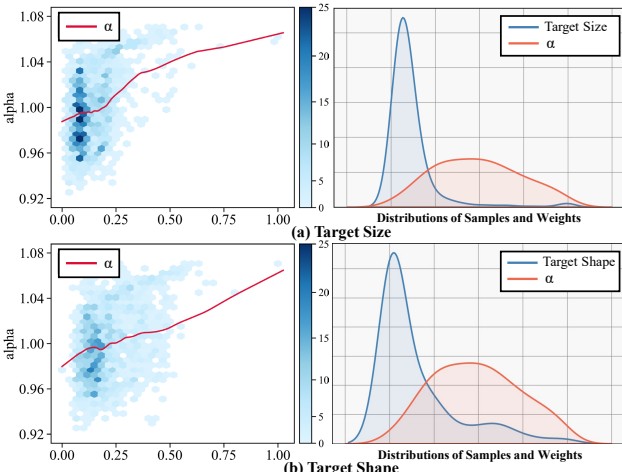

*Figure 5.* Analysis of the learned sample weight $\alpha$. The left plot shows the correlation of $\alpha$ with target size (small to large diameter) and target shape (near-circular to irregular), while the right plot shows the distribution of target shape, target size and $\alpha$.

*Table 3.* Ablation of core components in our framework. These results are obtained by ALCLNet.

| Model Variants | SIRST3 | | |
|---|---|---|---|
| | IoU↑ | nIoU↑ | $P_d$↑ |
| (A1) Naive point supervision | 8.07 | 17.51 | 78.02 |
| (A2) Diffusion only | 69.64 | 70.80 | 91.82 |
| (A3) Superpixel prior only | 68.64 | 69.41 | 93.16 |
| (A4) Diffusion + Superpixel (numerical) | 70.75 | 71.47 | 94.50 |
| (B1) Direct joint learning | 67.10 | 69.49 | 89.54 |
| (B2) w/o balance ($\alpha \equiv 1$) | 70.96 | 72.95 | 94.24 |
| (B3) w/o online label update | 71.52 | 71.26 | 93.43 |
| (B4) w/o dynamic aggregation ($\beta = 0$) | 67.74 | 68.60 | 92.23 |
| Proposed | 73.28 | 73.30 | 95.44 |

Fig. 3 provides a detailed qualitative comparison of our method against typical single-point supervision baselines (LESPS, MCLC, and PAL) on challenging examples from the SIRST3 dataset. The analysis highlights a clear pattern of robustness for our approach. In high-clutter environments where targets are adjacent to bright edges or complex structures (Rows 1 and 2), competing methods struggle significantly. A multi-target case is shown in the third row, where our method separates targets cleanly and avoids merging them into surrounding structures, while preserving target boundaries more effectively overall. A low-contrast multi-target scenario is shown in the last row, where our method achieves higher detection accuracy by successfully identifying faint targets that other methods often miss.

**Evaluation on three individual datasets.** We validated our framework's stability and robustness both qualitatively (Fig. 4) and quantitatively (Table 2) on individual datasets. In challenging high-clutter and low-SNR scenes, our method reliably localizes targets while suppressing noise. In contrast, baselines like LESPS, MCLC, and PAL suffer from unstable evolution, missed detections, and false alarms. This stability is quantitatively confirmed on scarce-data benchmarks, where our method consistently and significantly outperforms all competitors; for example, achieving over 20 percentage points of IoU improvement over LESPS.

### 4.3. Ablation Studies

**Efficiency of the proposed framework.** Table 3 presents an ablation analysis of the core components of our framework on SIRST3. Two obvious conclusions can be obtained. First, supervision quality is the primary bottleneck: Combining diffusion with the superpixel prior improves all metrics, suggesting complementary effects of diffusion expanding supervision, while the superpixel prior regularizes boundaries and suppresses label noise. Second, the training strategy is essential for turning better pseudo-labels into stable optimization: removing the bi-level dual-update, balance, online

label update, or dynamic aggregation (B1-B4) consistently degrades performance. Overall, these results support that effective pseudo-label construction and stable bi-level optimization are both necessary, and their combination leads to the best performance. More experiments about the sensitivity of hyperparameters are reported in Fig. 10.

**Verification of sample rebalancing.** To validate the meta-network's learned behavior, we conducted a statistical analysis of the weights $\alpha_i$ aggregated over 100 runs. As illustrated in Fig. 5, the weights correlate strongly with sample difficulty: challenging samples (*e.g.,* larger, irregular targets) receive higher $\alpha_i$, while easy samples (*e.g.,* small, compact targets) are assigned lower weights. Crucially, the weight density appears inversely correlated with the sample density; regions with high $\alpha$ density are precisely those where the sample density is low. The weighting branch automatically compensates for under-represented but challenging cases by upweighting them, while downweighting abundant cases.

**Validation of data usage.** We found that the learned weights $\alpha_i$ can also be used to guide data selection, reducing annotation and training costs. Experiments demonstrate that

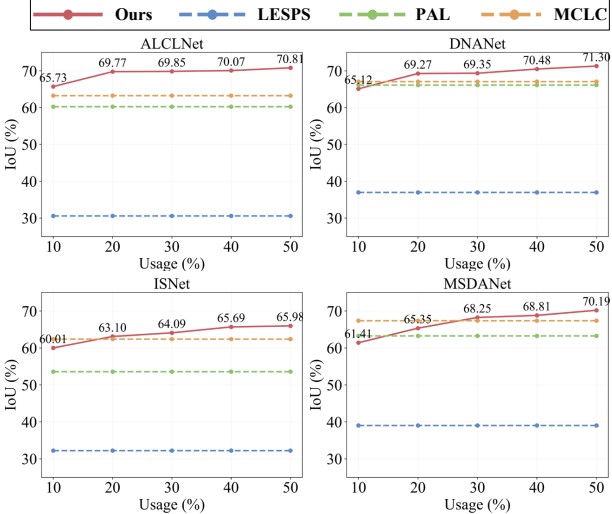

*Figure 6.* Data-efficiency on SIRST3 dataset across diverse networks, comparing Ours, LESPS, PAL, and MCLC under single-point supervision. Others are trained with full training datasets.

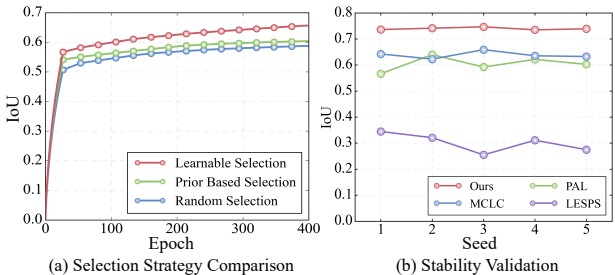

(a) Selection Strategy Comparison  (b) Stability Validation

*Figure 7.* Convergence of sample-selection strategy and stability validation on SIRST3 under diverse seeds.

a standard detector trained from scratch using only the top 30% of the sample subset, selected via $\alpha_i$ ranking, already surpasses the performance of baselines trained on 100% of the data, as shown in Fig. 6. This indicates that our bi-level optimization framework can effectively identify informative samples and safely discard redundant data. Fig. 7 (a) compares three data selection strategies under the same data budget. Our learned weights clearly outperform random and prior-based (hand-crafted difficulty) selection, showing more reliable identification of informative samples. Fig. 7 (b) reports results under five random seeds. Our method exhibits smaller variance than the baselines, demonstrating significantly improved training stability.

**Time efficiency of diffusion annotation.** We evaluated our proposed physics-induced diffusion annotation against existing single-point methods (COM and MCLC) in terms of both quality and efficiency. Moreover, as shown in Table 4, our pseudo-masks achieve an IoU of up to 68.02 against ground truth on the SIRST3 dataset, outperforming MCLC (66.98) and COM (15.62). As shown in Fig. 8, visual comparisons also confirm that while COM tends to

*Table 4.* Pseudo-mask generation on four ISTD datasets with Centroid and Coarse initialization, comparing COM, MCLC, and Ours, and reporting IoU and average time per target.

| Centroid | SIRST3 | SIRST-v1 | NUDT-SIRST | IRSTD-1k | Average Time↓ |
|---|---|---|---|---|---|
| | IoU↑ | IoU↑ | IoU↑ | IoU↑ | |
| COM (Li et al., 2024a) | 15.62 | 21.72 | 7.80 | 36.86 | 0.923s |
| MCLC (Li et al., 2023a) | 66.98 | 75.50 | 62.97 | 68.65 | 0.069s |
| Ours | 68.02 | 74.58 | 64.48 | 70.12 | 0.012s |
| Coarse | SIRST3 | SIRST-v1 | NUDT-SIRST | IRSTD-1k | Average Time↓ |
| | IoU↑ | IoU↑ | IoU↑ | IoU↑ | |
| COM (Li et al., 2024a) | 15.01 | 20.76 | 7.67 | 35.11 | 0.961s |
| MCLC (Li et al., 2023a) | 63.71 | 74.19 | 58.47 | 66.18 | 0.069s |
| Ours | 65.69 | 73.78 | 59.15 | 68.42 | 0.013s |

*Table 5.* Pseudo-mask generation on four ISTD datasets with Centroid and Coarse initialization, comparing SAM, SAM2, and Ours, reporting IoU and average time per target.

| Centroid | SIRST3 | SIRST-v1 | NUDT-SIRST | IRSTD-1k | Average Time↓ |
|---|---|---|---|---|---|
| | IoU↑ | IoU↑ | IoU↑ | IoU↑ | |
| SAM (Kirillov et al., 2023) | 44.98 | 55.39 | 41.54 | 45.41 | 0.105s |
| SAM2 (Ravi et al., 2025) | 47.39 | 57.01 | 47.91 | 43.31 | 0.156s |
| Ours | 68.02 | 74.58 | 64.48 | 70.12 | 0.012s |
| Coarse | SIRST3 | SIRST-v1 | NUDT-SIRST | IRSTD-1k | Average Time↓ |
| | IoU↑ | IoU↑ | IoU↑ | IoU↑ | |
| SAM (Kirillov et al., 2023) | 6.03 | 4.41 | 7.98 | 4.13 | 0.135s |
| SAM2 (Ravi et al., 2025) | 47.14 | 57.06 | 47.67 | 42.95 | 0.149s |
| Ours | 65.69 | 73.78 | 59.15 | 68.42 | 0.013s |

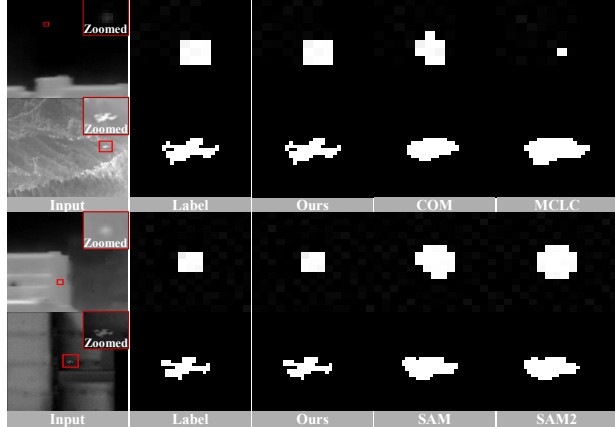

*Figure 8.* Qualitative comparison of small target detection on the SIRST3 dataset: Ours vs. COM, MCLC, SAM, and SAM2.

under-segment and MCLC often over-grows into the background, our method better maintains clean target boundaries. In terms of efficiency, our method requires only 0.012 seconds per target on a CPU, which is approximately five times faster than MCLC (0.069s) and nearly two orders of magnitude faster than COM (0.923s). This high efficiency (a five-fold speedup over MCLC) is crucial: it makes the diffusion module lightweight enough to be embedded within the bi-level optimization loop. The training efficiency is reported in Table 6. We compared our proposed physics-diffusion annotation with foundation models SAM/SAM2 in Table 5. Visually, SAM/SAM2 perform poorly on infrared small targets, often either missing them or causing severe over-segmentation, as shown in Fig. 8. In contrast, our method generates compact masks that better follow the target's true thermal footprint. In terms of efficiency, our dif-

fusion module is also significantly faster than SAM/SAM2. This demonstrates that for single-point supervised ISTD, a dedicated physics-guided framework remains superior to generic foundation models in both accuracy and efficiency.

## 5. Conclusion

In this paper, we proposed a bi-level dual-update framework built upon single-point supervision to address annotation costs and sample imbalance jointly. First, we introduced a physics-induced annotation strategy. Second, we addressed the joint optimization of the detector, sample importance weights, and annotation quality. Extensive experiments demonstrated that our approach is highly efficient, achieving superior detection accuracy.

## Impact Statement

This paper presents work whose goal is to advance the field of Machine Learning. There are many potential societal consequences of our work, none which we feel must be specifically highlighted here.

## Acknowledgements

This work is partially supported by the National Natural Science Foundation of China (Nos.U22B2052, 624B2033), Central Guidance for Local Science and Technology Development Fund (Youth Science Fund Project, Category A, No.2025JH6/101100001), the Distinguished Young Scholars Funds of Dalian (No.2024RJ002), and the Fundamental Research Funds for the Central Universities.

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

# A. Appendix

This appendix is organized as follows: First, we provide a comprehensive description of the architectural details of the meta-classifier in Appendix A.1. Second, we illustrate the diffusion process used for label generation in Appendix A.2. We also elaborate on the details of the whole algorithm in Appendix A.3. Finally, we present additional experiments to further validate the effectiveness of our proposed method, including the sensitivity of hyperparameters, comparison of training procedures, extension to other modalities, and more objective and subjective comparison in Appendix A.4.

### A.1. Architecture Details of Meta-Classifier

In Eq. (1) of the main paper, the sample-wise loss weights $\alpha = \{\alpha_i\}$ are treated as hyperparameters that modulate the inner-level training loss, while being refined by the outer-level validation objective. This section provides the architectural details of the meta-classifier that predicts $\alpha_i$, as well as the construction of its input features.

**Meta-classifier overview.** Let $\mathcal{N}(x_i; \omega)$ denote the detector with parameters $\omega$, and let $y_i = \mathcal{N}(x_i; \omega) \in \mathbb{R}^{H \times W}$ be the logit map for sample $i$. We write $p_i = \sigma(y_i) \in [0, 1]^{H \times W}$ for the corresponding probability map, and $\hat{y}_i$ for the current pseudo-label used in the inner-level loss, consistent with Eq. (1). The meta-classifier is implemented as a lightweight MLP $g_\psi(\cdot)$ with parameters $\psi$, which takes a sample-wise feature vector $h_i$ and outputs a scalar logit $\ell_i = g_\psi(h_i)$. The loss weight $\alpha_i$ is obtained by a sigmoid followed by batch-wise normalization:

$$\alpha_i = \frac{\sigma(\ell_i)}{\frac{1}{B} \sum_{j=1}^{B} \sigma(\ell_j)}, \tag{5}$$

where $B$ is the batch size.

**Network architecture.** We use a two-layer MLP with ReLU as the meta-classifier $g_\psi$:

$$g_\psi(h_i) = W_2 \, \phi(W_1 h_i + b_1) + b_2. \tag{6}$$

The meta-classifier is optimized only in the outer loop (Adam, learning rate $1 \times 10^{-3}$), while detector parameters $\omega$ are updated in the inner loop.

### A.2. Diffusion Procedure for Label Generation

Given an infrared image and sparse point annotations, we generate a pseudo mask by (i) building a complexity-aware diffusivity field that suppresses diffusion across complex/background structures, (ii) propagating heat from the annotated point(s) to obtain a thermal field, (iii) constraining the support using a compact region prior, and (iv) producing a binary mask via a parameter-free thresholding step. We implement this design in two forms: an offline annotator that operates on the input image to produce on-disk pseudo masks, and a differentiable surrogate that operates on network predictions to enable outer-loop updates. Both share the same components but use different solvers.

**Offline physics-induced diffusion annotator (Fig. 9 a–j).** For each annotated target point, we crop a local patch and perform:

- **(a) Pre-processing.** Normalize the infrared intensity and apply a standard local contrast enhancement to obtain an intensity map $I \in [0, 1]$.

- **(b–d) Complexity-aware diffusivity.** Compute a complexity map $C$ from simple local statistics (e.g., gradient magnitude and local entropy), and map it to a spatially varying diffusion coefficient $D_\theta$ such that diffusion is strong in homogeneous regions and weak near complex/background structures. This yields Fig. 9(b–d).

- **(f–h) Diffusion solver (random-walk heat propagation).** Starting from the annotated point as the heat source, we simulate heat propagation using a random-walk / particle diffusion solver guided by $D_\theta$ and accumulate visit counts into a normalized thermal field $u^{(K)} \in [0, 1]$. Intermediate fields correspond to Fig. 9(f–h).

- **(e,i) Compact region prior and fusion.** To avoid over-diffusion, we estimate a compact region prior $C_p$ around the source point (e.g., via superpixel segmentation or an equivalent region extractor), and fuse it with the thermal field:

$$\hat{y}^{\text{soft}} = \rho \, u^{(K)} + (1 - \rho) \, C_p,$$

which matches Fig. 9(e,i). Masks from multiple points are fused by union/maximum to form the full-image soft mask.

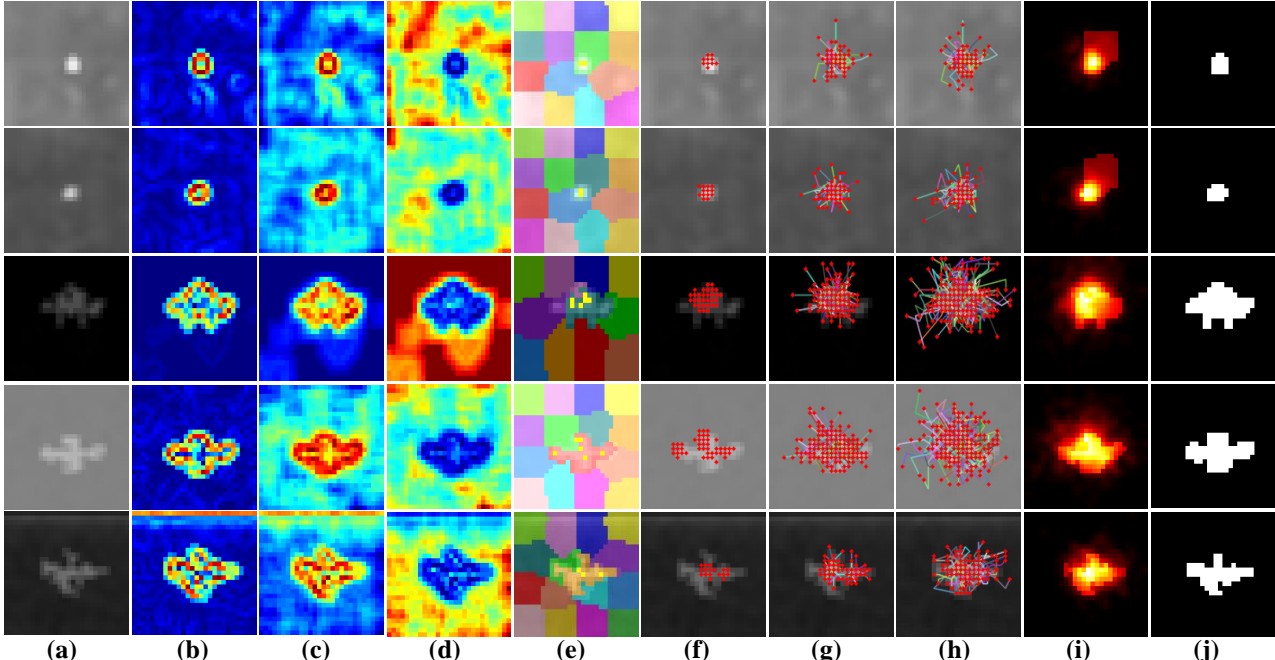

|  |  |  |  |  |  |  |  |  |  |
|---|---|---|---|---|---|---|---|---|---|
| **(a)** | **(b)** | **(c)** | **(d)** | **(e)** | **(f)** | **(g)** | **(h)** | **(i)** | **(j)** |

*Figure 9.* Illustration of the offline physics-induced diffusion annotator. (a) Cropped input patch $x$. (b) Gradient magnitude map $\|\nabla I(a,b)\|$. (c) Normalized local entropy $H(a,b)$ used in the complexity measure $C(a,b)$. (d) Image-adaptive scalar diffusion coefficient $D_\theta(a,b)$ derived from $C(a,b)$. (e) Superpixel prior $C_{p_i}(a,b)$ and selected diffusion source points (yellow pixels). (f)–(h) Intermediate thermal fields $u^{(k)}$ during the random-walk diffusion process. (i) Fused soft field $\hat{y}_i^{\text{soft}}(\theta) = \rho\, u^{(K)} + (1-\rho)\, C_{p_i}(a,b)$, combining the particle history heatmap with the superpixel prior. (j) Final physics-induced pseudo mask $\hat{y}_i^{\text{phys}}$ after thresholding.

- **(j) Binarization.** We apply a parameter-free global thresholding (Otsu) on $\hat{y}^{\text{soft}}$ to obtain the final offline pseudo mask $\hat{y}^{\text{phys}}$ (Fig. 9j).

**Differentiable diffusion surrogate.** Direct backpropagation through the random-walk solver is expensive. We therefore implement a differentiable surrogate $\mathcal{M}(y;\theta)$ that mirrors the above components but operates on the detector prediction. Let $P = \sigma(y) \in [0,1]^{H \times W}$ be the probability map.

- Compute $C(P)$ using simple local statistics on $P$ (gradient/entropy proxies), and produce $D_\theta(P)$ using the same monotone decreasing mapping as the offline annotator.

- Approximate heat propagation with a small number of explicit diffusion steps (discrete anisotropic diffusion):

$$U^{(0)} = P, \qquad U^{(k+1)} = U^{(k)} + D_\theta(P) \odot \Delta U^{(k)}, \ \ k = 0, \ldots, K_s - 1,$$

with clamping to $[0,1]$ after each step.

- Replace the offline region prior $C_p$ with a prediction-derived source prior $S$ (e.g., a sparse high-confidence seed map from $P$), and fuse it with the diffused field:

$$F = \rho\, U^{(K_s)} + (1-\rho)\, S.$$

- Output a soft mask via a squashing function:

$$\mathcal{M}(y;\theta) = \sigma(F),$$

which is differentiable w.r.t. $\theta$.

---

**Algorithm 1** Bi-level Dual-update Framework

---

**Require:** Training set $\mathcal{D}_{tr} = \{(x_i, p_i)\}_{i=1}^{N}$ (image $x_i$, single point $p_i$); validation set $\mathcal{D}_{val}$; detector $\mathcal{N}(\cdot; \omega)$; meta-classifier $g_\psi(\cdot)$; differentiable diffusion surrogate $\mathcal{M}(\cdot; \theta)$. Hyperparameters: diffusion steps $K$, step size $\tau$, aggregation coefficient $\beta$, learning rates $\eta_\omega, \eta_\alpha, \eta_\theta$.

**Ensure:** Detector parameters $\omega$.

1: **(Offline) Physics-induced diffusion annotation** *(Eq. (2)–(4))*
2: **for** each $(x_i, p_i) \in \mathcal{D}_{tr} \cup \mathcal{D}_{val}$ **do**
3:     Construct image-adaptive diffusion tensor $D_\theta(a, b)$ and Laplacian operator $L_{D_\theta}$ from $x_i$;
4:     Initialize thermal potential $u^{(0)} = \delta_{p_i}$
5:     **for** $k = 0$ **to** $K - 1$ **do**
6:         $u^{(k+1)} = (I - \tau L_{D_\theta})u^{(k)}$; *(Eq. (3))*
7:     **end for**
8:     Obtain superpixel prior $C_{p_i}(a, b)$ (superpixel region containing $p_i$);
9:     Fuse two sources to form soft pseudo-label:
10:        $\hat{y}_i(\theta) = \rho \cdot u^{(K)}(a, b) + (1 - \rho) \cdot C_{p_i}(a, b)$, then normalize to $[0, 1]$; *(Eq. (4))*
11: **end for**
12: % $\rho \in [0, 1]$ *is learnable;* $\theta$ *parameterizes* $D_\theta$ *and* $\rho$ .
13: **Initialize** $\omega^0, \psi^0, \theta^0$
14: Warm-start the detection network for 80 epochs.
15: **for** outer iteration $t = 0, 1, 2, \dots$ **do**
16:     **(1) Inner-level update of detector with dynamic aggregation**
17:     Set $\omega^{t,0} \leftarrow \omega^t$;
18:     **for** $n = 0$ **to** $T - 1$ **do**
19:         Sample mini-batches $\mathcal{B}_{tr} \subset \mathcal{D}_{tr}$ and $\mathcal{B}_{val} \subset \mathcal{D}_{val}$;
20:         Compute weighted training loss:
21:         $L_{tr}(\omega^{t,n}, \alpha^t, \theta^t) = \sum\limits_{i \in \mathcal{B}_{tr}} \alpha_i^t \, \mathcal{L}_{tr}(\mathcal{N}(x_i; \omega^{t,n}), \hat{y}_i(\theta^t))$;
22:         Compute validation loss:
23:         $L_{val}(\omega^{t,n}, \alpha^t, \theta^t) = \sum\limits_{j \in \mathcal{B}_{val}} \mathcal{L}_{val}(\mathcal{N}(x_j; \omega^{t,n}), \hat{y}_j(\theta^t))$;
24:         Update detector parameters (dynamic aggregated solution):
25:         $\omega^{t,n+1} = \omega^{t,n} - \eta_\omega\Big(\nabla_\omega L_{tr}(\omega^{t,n}, \alpha^t, \theta^t) + \beta\,\nabla_\omega L_{val}(\omega^{t,n}, \alpha^t, \theta^t)\Big)$;
26:     **end for**
27:     Set $\omega^{t+1} \leftarrow \omega^{t,T}$;
28:     **(2) Outer-level update of sample weights** *(Eq. (1), one-step update)*
29:     Update meta-classifier parameters $\psi$ through validation objective:
30:     $\alpha^{t+1} = \alpha^t - \eta_\alpha \nabla_\alpha L_{val}(\omega^{t+1}, \alpha^t, \theta^t)$
31:     **(3) Refine diffusion parameters via surrogate alignment** *(Sec. 3.3)*
32:     For each $i \in \mathcal{B}_{tr}$, let $y_i = \mathcal{N}(x_i; \omega^{t+1})$ and compute $\tilde{y}_i = \mathcal{M}(y_i; \theta^t)$;
33:     Update diffusion parameters by aligning surrogate diffusion to offline pseudo-labels:
34:     $\theta^{t+1} = \theta^t - \eta_\theta \nabla_\theta \sum\limits_{i \in \mathcal{B}_{tr}} \Big(\mathcal{L}_{val}\big(\mathcal{M}(y_i; \boldsymbol{\theta}), \hat{y}_i\big) + \mathcal{L}_{val}\big(\mathcal{M}(y_i; \boldsymbol{\theta}), y_i\big)\Big)$;
35: **end for**
36: **return** $\omega^*$.

---

### A.3. Dynamic Aggregated Solution

The overall framework is summarized in Alg. 1. We first generate soft pseudo-labels $\hat{y}_i(\theta)$ offline by fusing a physics-induced anisotropic diffusion field with a superpixel prior, where the diffusion is run for $K = 30$ steps with step size $\tau = 0.8$ on average. Then, at each outer iteration, we update the detector parameters $\omega$ for $T = 4$ inner steps using a dynamic aggregated gradient that combines the training loss and the validation loss, where $\beta = 0.8$ controls the strength of validation guidance and the detector learning rate is $\eta_\omega = 1 \times 10^{-3}$. After obtaining $\omega^{t+1}$, we perform lightweight outer updates: (i) update the sample weights $\alpha$ (via the meta-classifier) with learning rate $\eta_\alpha = 1 \times 10^{-3}$, and (ii) refine diffusion parameters $\theta$ by aligning the differentiable diffusion surrogate $\mathcal{M}(\cdot; \theta)$ with the offline pseudo-labels and the current detector outputs, using a learning rate of $\eta_\theta = 1 \times 10^{-2}$ and updating it every 20 epochs. We warm-start the detector for 80 epochs before the bi-level optimization. Repeating these steps yields an efficient closed-loop optimization between supervision generation and detector learning.

### A.4. More Experimental Results

**Sensitivity analysis of parameters.** We evaluate the training performance under different settings of $\beta$ and $\rho$, as well as the learning rate and update frequency used for mask refinement. The results are summarized in Fig. 10. For the dynamic aggregation coefficient, extensive experiments show that $\beta = 0.8$ yields the optimal trade-off between training supervision and validation guidance. $\rho$ is treated as a learnable parameter. Empirically, after repeated runs and statistical analysis across multiple trainings, the learned $\rho$ consistently converges to around $0.8$, at which the model achieves the best performance, indicating an effective fusion between the diffusion and the superpixel prior. Regarding mask refinement, we find that updating pseudo-labels every 20 epochs with a refinement learning rate of $1 \times 10^{-2}$ provides the most reliable improvements, and we adopt these settings as the default configuration.

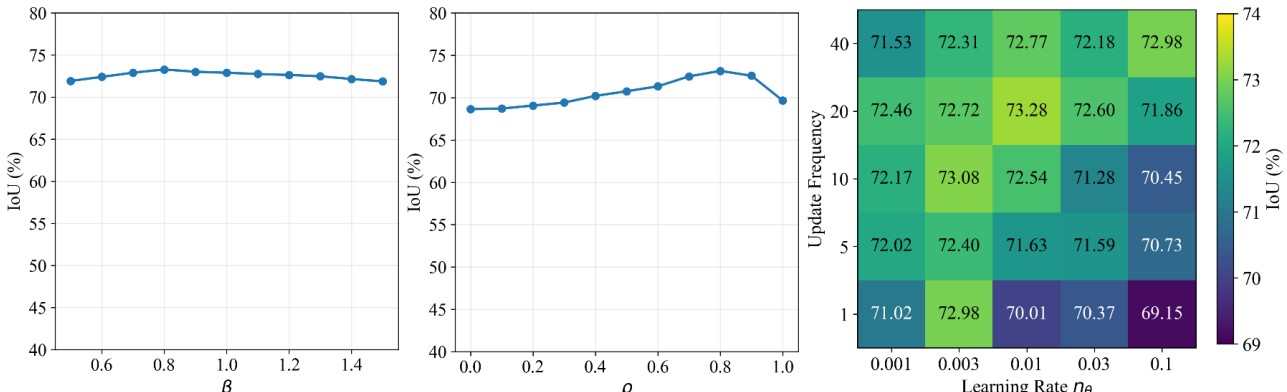

*Figure 10.* Sensitivity analysis of $\beta$, $\rho$, label update frequency (epochs) and learning rate $\eta_\theta$.

**Training time comparison.** To further examine the practical efficiency of different point-supervised training paradigms, we compare the average per-epoch training time for Full, LESPS, PAL, MCLC, and Ours. All experiments are conducted with ALCLNet trained on SIRST3 on a single NVIDIA A40 GPU with CUDA 11.8. For a fair comparison, we only report the core-stage time of each method: for LESPS and PAL, we measure epochs after pseudo-label evolution starts; for Ours, we measure epochs after the bi-level optimization starts. As shown in Table 6, LESPS introduces the largest time overhead compared with Full, mainly because it relies on intensive pseudo-label evolution throughout training, which repeatedly updates and propagates masks and thus increases computational cost. In contrast, PAL is faster than Full: although it also performs label evolution, its curriculum learning strategy effectively reduces the number of samples actively involved at different stages, leading to a lower average training time per epoch. MCLC exhibits a training cost comparable to Full, since it is an offline strategy and does not incur substantial additional online refinement during training. Our method adopts a bi-level optimization formulation; however, the actual training time is only marginally higher than Full. This is primarily attributed to our proposed dynamic aggregation strategy, which provides efficient validation-guided updates without expensive inner-loop unrolling. We then report test IoU results over 400 epochs for all methods. Our approach achieves the best test IoU among all compared methods, demonstrating a favorable balance between effectiveness and efficiency and supporting its practical usability.

*Table 6.* Efficiency comparison. We report supervision type, whether label evolution and curriculum learning are used, whether bi-level optimization (proposed) is adopted, and the average per-epoch training time (seconds, averaged over epochs) and the test IoU (%).

| Method | Label Type | Label Update | Curriculum Learning | Bi-level Optimization | Avg. Epoch Time ↓ | Test IoU ↑ |
|--------|-----------|--------------|---------------------|-----------------------|-------------------|------------|
| Full | Full | ✗ | ✗ | ✗ | 61.6471 | 82.12 |
| LESPS | Point | LESPS evolution | ✗ | ✗ | 68.2024 | 30.59 |
| PAL | Point | LESPS evolution + Degeneration | ✓ | ✗ | 53.0894 | 60.26 |
| MCLC | Point | ✗ | ✗ | ✗ | 62.3093 | 63.27 |
| Ours | Point | Learnable diffusion | ✗ | ✓ | 64.6010 | 73.28 |

**Extension to other heterogeneous modalities.** In Fig. 11, we demonstrate the generalization of our point diffusion annotation across multiple modalities, including visible-band remote sensing, SAR, and sonar. In detail, SAM2 easily fails to effectively detect the targets across these heterogeneous modalities, leaving large areas of the objects undetected or generating abundant false-positive regions. In contrast, our scheme achieves superior detection performance, effectively identifying and segmenting targets in all three modalities. Coupled with sample balancing and online label refinement, our framework can be uniformly applied to address the detection of small targets in different modalities in future work.

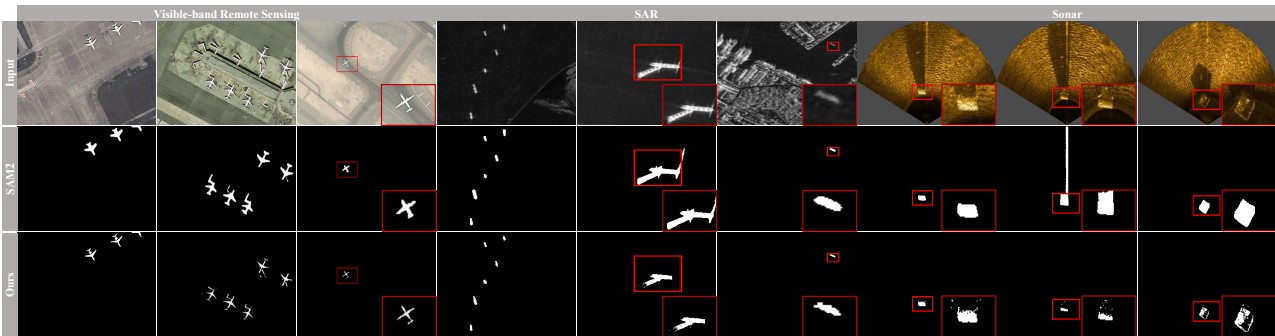

*Figure 11.* Generalization of point diffusion across different modalities: visible-band remote sensing, SAR, and sonar.

**Feature definitions and supplementary plots.** For the analysis in Fig. 5 of the main paper and the additional plots in the supplement (Fig. 12), we use five image-level descriptors computed from the final physics-induced pseudo mask $\hat{y}_i^{\text{phys}} \in \{0, 1\}^{H \times W}$ of image $i$. Let $\Omega = \{1, \ldots, H\} \times \{1, \ldots, W\}$ denote the image lattice and let

$$\mathcal{R} = \{R_k \subset \Omega \mid k = 1, \ldots, K\}$$

be the set of connected foreground components obtained from $\hat{y}_i^{\text{phys}}$. We select the largest component $R^\star = \arg\max_k |R_k|$ and use its area and perimeter as the basis of the size/shape descriptors.

- **Target size.** The target size is measured by the equivalent diameter of the largest connected component. Let $A^\star = |R^\star|$ denote its area (number of foreground pixels). The equivalent diameter is defined as the diameter of a circle with the same area:

$$d_{\text{size}} = 2\sqrt{\frac{A^\star}{\pi}}. \tag{7}$$

- **Target shape.** To characterize the compactness of the largest component, we compute its perimeter $P^\star$ (8-connected) and define the shape descriptor as

$$c_{\text{shape}} = \frac{(P^\star)^2}{4\pi A^\star}, \tag{8}$$

which equals 1 for a perfect disk and increases as the region becomes more elongated or irregular.

- **Target area ratio.** Besides the absolute size, we also use the fraction of image area occupied by the target foreground:

$$r_{\text{area}} = \frac{1}{HW} \sum_{(a,b) \in \Omega} \hat{y}_i^{\text{phys}}(a, b). \tag{9}$$

This descriptor reflects how large the entire target region is relative to the full field of view.

- **Target edge density.** To capture structural complexity inside the target, we apply a Canny edge detector to the input image (with fixed thresholds in the implementation) and obtain a binary edge map $E(a, b) \in \{0, 1\}$. The edge density within the target is defined as

$$d_{\text{edge}} = \frac{\sum_{(a,b)\in\Omega} E(a, b)\, \hat{y}_i^{\text{phys}}(a, b)}{\sum_{(a,b)\in\Omega} \hat{y}_i^{\text{phys}}(a, b) + \varepsilon}, \tag{10}$$

  i.e., the proportion of pixels inside the foreground mask that are detected as edges. A larger value indicates a more textured or structurally complex target interior.

- **Centroid distance.** Finally, we measure how far the target is located from the image center. Let $(x_c, y_c)$ be the centroid of the largest component $R^\star$, computed in continuous coordinates, and let the image center be

$$(x_0, y_0) = \left( \frac{W - 1}{2}, \frac{H - 1}{2} \right).$$

We define the centroid distance as

$$d_{\text{cen}} = \frac{\sqrt{(x_c - x_0)^2 + (y_c - y_0)^2}}{\sqrt{(W - 1)^2 + (H - 1)^2}}, \tag{11}$$

which lies in $[0, 1]$ and equals 0 when the target is centered in the image and approaches 1 when it is located near the image corners.

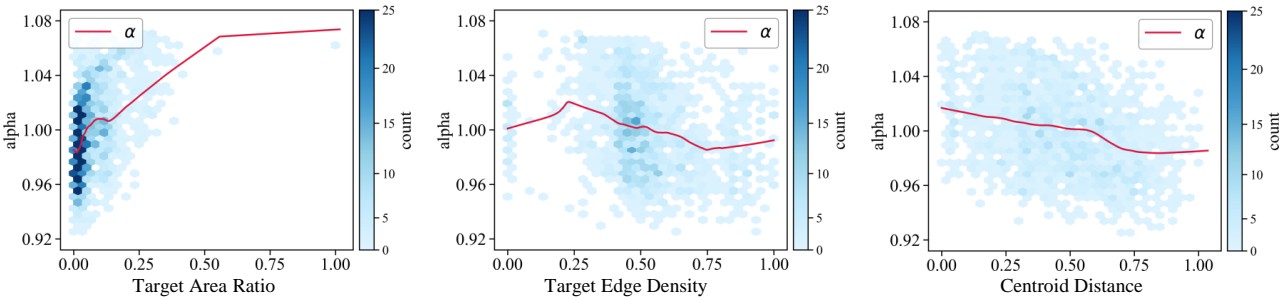

*Figure 12.* Analysis of the correlation of $\alpha$ with target area ratio, target edge density, and centroid distance.

**Additional quantitative and qualitative comparisons.** In addition to the results presented in the main paper, we include a number of further quantitative and qualitative comparisons with baseline methods. These extended results are provided in Table 7 and Fig. 13, 14, and 15.

*Table 7.* Additional quantitative results. Models are trained on three individual datasets (SIRST-v1, NUDT-SIRST, IRSTD-1k). We report four metrics: IoU (%), nIoU (%), $P_d$ (%), and $F_a$ ($10^{-6}$). "-" indicates that training is unstable.

| Methods | Strategy | SIRST-v1 | | | | NUDT-SIRST | | | | IRSTD-1k | | | |
|---|---|---|---|---|---|---|---|---|---|---|---|---|---|
| | | IoU↑ | nIoU↑ | $P_d$↑ | $F_a$↓ | IoU↑ | nIoU↑ | $P_d$↑ | $F_a$↓ | IoU↑ | nIoU↑ | $P_d$↑ | $F_a$↓ |
| ACMNet (Dai et al., 2021a) | Full | 59.49 | 57.39 | 86.31 | 38.42 | 64.21 | 62.20 | 96.83 | 9.49 | 59.70 | 54.91 | 90.24 | 24.65 |
| | LESPS | - | - | - | - | 36.98 | 35.87 | 79.79 | 22.75 | 29.35 | 38.07 | 76.43 | 34.50 |
| | PAL | - | - | - | - | 53.09 | 51.44 | 93.86 | 19.72 | 46.73 | 44.57 | 87.21 | 29.68 |
| | MCLC | 62.30 | 62.20 | 89.73 | 60.23 | 56.64 | 55.97 | 91.96 | 34.01 | 56.38 | 51.15 | 82.85 | 35.74 |
| | Ours | 65.16 | 63.53 | 98.15 | 11.31 | 57.91 | 59.82 | 95.06 | 15.72 | 54.99 | 51.47 | 88.55 | 29.55 |
| MLCLNet (Yu et al., 2022a) | Full | 75.80 | 77.48 | 95.82 | 25.59 | 93.56 | 93.38 | 98.73 | 2.76 | 64.31 | 62.85 | 91.25 | 21.83 |
| | LESPS | 54.15 | 64.17 | 90.49 | 36.50 | 38.89 | 46.16 | 86.77 | 32.02 | 36.54 | 50.73 | 80.47 | 29.58 |
| | PAL | 66.65 | 70.86 | 92.02 | 34.85 | 72.66 | 73.39 | 96.08 | 26.13 | 49.86 | 56.59 | 89.23 | 28.58 |
| | MCLC | 64.29 | 69.73 | 96.96 | 22.09 | 69.82 | 71.46 | 95.24 | 30.43 | 53.59 | 56.08 | 84.85 | 25.24 |
| | Ours | 67.58 | 71.62 | 98.15 | 20.11 | 73.85 | 82.58 | 96.22 | 23.35 | 60.89 | 58.43 | 91.92 | 13.97 |
| ISNet (Zhang et al., 2022b) | Full | 72.90 | 74.41 | 93.54 | 42.88 | 86.11 | 87.51 | 95.56 | 22.75 | 63.02 | 61.40 | 88.89 | 21.71 |
| | LESPS | - | - | - | - | 39.15 | 48.02 | 83.07 | 42.47 | 24.96 | 25.76 | 70.76 | 36.08 |
| | PAL | 36.56 | 48.83 | 73.38 | 32.86 | 52.48 | 58.44 | 89.52 | 43.52 | 31.00 | 42.20 | 74.07 | 33.63 |
| | MCLC | 60.72 | 64.10 | 90.49 | 85.82 | 68.19 | 71.31 | 92.70 | 36.70 | 63.01 | 58.67 | 87.88 | 23.47 |
| | Ours | 70.79 | 70.18 | 98.15 | 26.21 | 73.13 | 81.51 | 95.93 | 36.20 | 62.53 | 59.82 | 88.78 | 23.00 |
| GGLNet (Zhao et al., 2023) | Full | 76.48 | 76.90 | 96.96 | 25.38 | 92.59 | 93.11 | 99.15 | 4.78 | 65.62 | 65.04 | 92.93 | 29.44 |
| | LESPS | 54.83 | 57.07 | 85.17 | 25.71 | 43.61 | 49.52 | 75.45 | 30.31 | 36.29 | 51.40 | 79.80 | 25.41 |
| | PAL | 62.50 | 64.98 | 90.49 | 34.10 | 58.84 | 64.22 | 93.54 | 30.24 | 60.91 | 59.04 | 88.55 | 23.95 |
| | MCLC | 73.34 | 73.42 | 95.44 | 24.83 | 70.66 | 73.33 | 95.24 | 30.70 | 63.51 | 60.03 | 84.18 | 32.76 |
| | Ours | 74.47 | 73.71 | 99.07 | 13.10 | 80.47 | 86.44 | 98.26 | 9.69 | 64.55 | 60.65 | 88.78 | 15.49 |

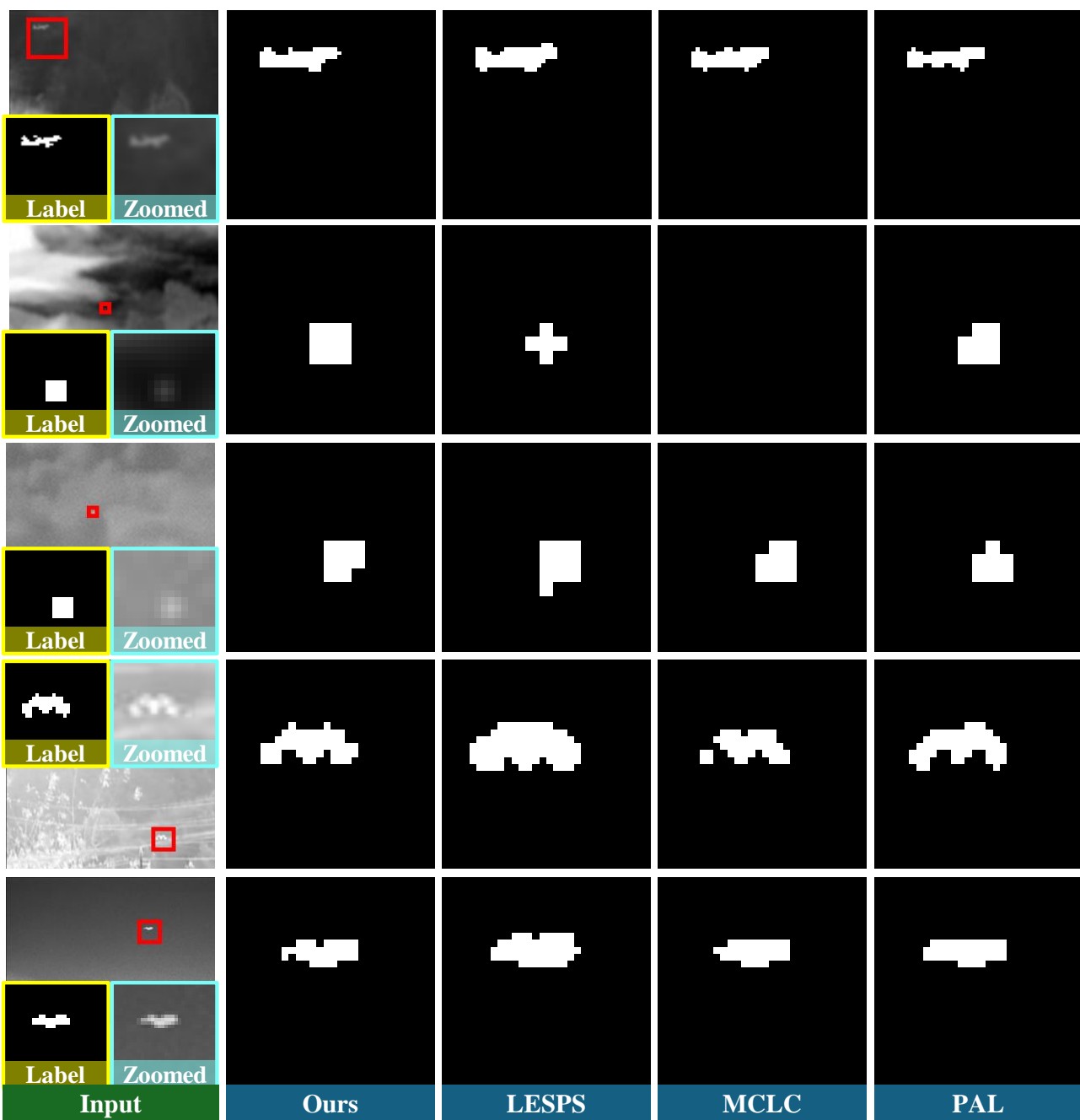

*Figure 13.* Additional qualitative comparisons, trained with ALCLNet.

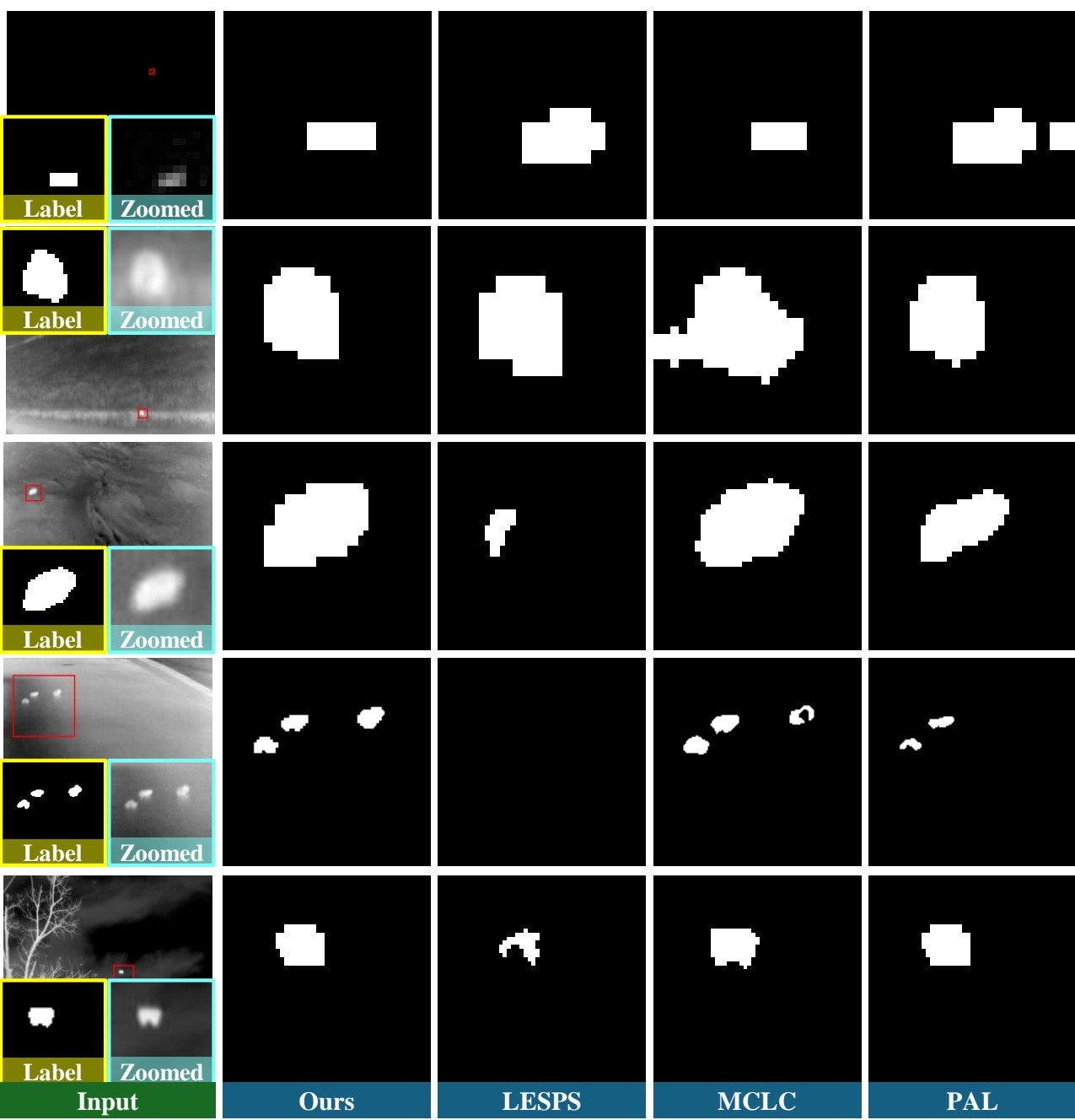

*Figure 14.* Additional qualitative comparisons, trained with DNANet.

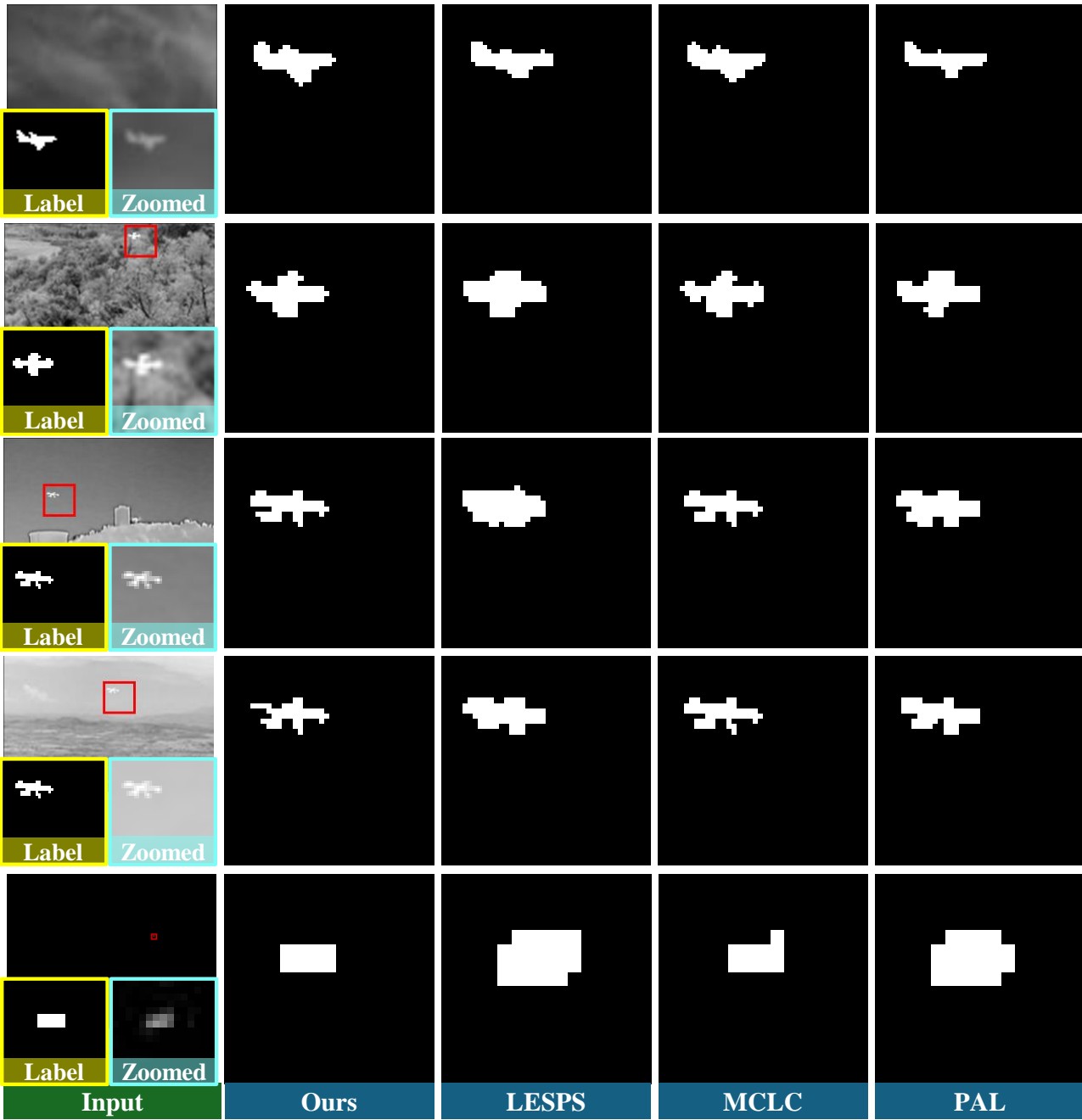

*Figure 15.* Additional qualitative comparisons, trained with MSDANet.

