# OpenReview forum: "Diffuse to Detect: Bi-Level Sample Rebalancing with Pseudo-Label Diffusion for Point-Supervised Infrared Small-Target Detection"
_ICML.cc/2026/Conference — ICML 2026 spotlight_

### Official Review · Reviewer_YyzY · 2026-03-07

**Soundness:** 3
**Presentation:** 3
**Significance:** 2
**Originality:** 2
**Overall Recommendation:** 4
**Confidence:** 4

**Summary:**

This paper addresses point-supervised infrared small target detection, where single-point annotations are expanded into pseudo-labels for training. The authors identify two main challenges: unstable pseudo-label evolution in low-SNR infrared imagery and severe sample-distribution imbalance. To address these issues, they propose a bi-level dual-update optimization framework that jointly updates detector parameters, sample weights, and pseudo-label generation. A physics-induced diffusion model expands point annotations into pseudo-masks based on thermal diffusion principles, while a meta-classifier predicts sample-wise loss weights to mitigate imbalance. A differentiable diffusion module further refines pseudo-labels using detector feedback.

**Compliance With Llm Reviewing Policy:**

Affirmed.

**Final Justification:**

My concerns have been adequately addressed.

**Key Questions For Authors:**

Please response to the weaknesses if they are misinterpreted. I will reassess the paper based on the response.

**Limitations:**

There is no limitation discussed.

**Strengths And Weaknesses:**

Strengths:
1. The problem formulation and motivation are generally clear. The method is structured into understandable components (physics-induced annotation, bi-level optimization, and dynamic aggregation), and the technical design is logically connected to the identified bottlenecks.
2. The experiments are extensive. The method is incorporated with many detectors and evaluated on multiple datasets. Experiments show improved detection performance.

Weaknesses:
1. The technical narrative sometimes overstates the physical interpretation of the diffusion model. For infrared small targets, the object often occupies only a few pixels. The actual heat diffusion in the physical world rarely corresponds to the spatial intensity distribution in the image. It seems to me that the method is essentially spreading the label to similar pixels around the point annotation, and the authors find a physical concept to make it complicated.
2. Many core components resemble existing ideas: diffusion-based label propagation, meta-learning for sample weighting, and bi-level optimization. The contribution largely lies in combining these elements, which somewhat limits the novelty.
3. Point-supervised detection methods for general targets, such as Wholly-WOOD, PointOBB, Point2RBox-v2, are highly related, but not discussed in the related work.
4. The authors put Impact Statement in the Page 9, which does not conform with "The paper can be up to eight pages long, followed by any number of pages for references and appendices" rule.

---

> ### Author Rebuttal · Authors · 2026-03-30
>
> We sincerely thank the reviewer for the careful and constructive assessment. We appreciate the recognition of the clear motivation, structured technical design, and extensive experiments. We respond to the main concerns below.
>
> (1) Clarification of the physical interpretation.
> We thank the reviewer for this important comment. We agree that our current wording may make the physical interpretation appear stronger than intended. Our goal is not to claim that the image-space diffusion process exactly matches real-world heat diffusion for infrared small targets. Instead, the module should be understood as a **physics-inspired** local propagation prior for point-supervised pseudo-label generation. Its role is to keep pseudo-label expansion local, adaptive to surrounding image structures, and resistant to noisy background propagation. This broader interpretation is also consistent with its generalization across heterogeneous modalities such as visible-band remote sensing, SAR, and sonar in Fig. 11. We will revise the wording accordingly to avoid overstating the physical claim.
>
> (2) Novelty and contribution positioning.
> We acknowledge that the contribution of our work could be positioned more clearly in the current version. We believe the main contribution lies in addressing **two strongly coupled bottlenecks** in point-supervised infrared small target detection, namely unstable pseudo-label evolution and severe sample-distribution imbalance, within a **unified dual-update bi-level optimization framework**.
>
> Rather than treating pseudo-label generation, detector learning, and sample balancing as separate procedures, our method couples them through a closed-loop interaction: detector feedback refines pseudo-label generation, while validation-guided reweighting reshapes the effective training distribution, and both updates further affect subsequent detector optimization. This coupled design is reflected not only in the formulation, but also in the empirical evidence already presented in the paper.
>
>  In particular, **Table 3** shows that removing the bi-level dual-update or other key components, such as online label update and dynamic aggregation, leads to clear performance drops; **Fig. 5** shows that the learned weights are meaningfully aligned with sample difficulty; **Fig. 7(b)** shows improved stability under different random seeds; and the consistent gains in **Tables 1–2** across multiple detectors and datasets further suggest that the benefit comes from the **coordinated interaction** within the proposed framework, **rather than from isolated procedures applied independently**. We will revise the paper to make this contribution positioning more explicit.
>
> (3) Missing related works.
> We thank the reviewer for pointing out these relevant works. We agree that Wholly-WOOD, PointOBB, and Point2RBox-v2 are relevant references in the broader literature on point-supervised detection. Wholly-WOOD studies unified weakly supervised oriented object detection with Point/HBox/RBox annotations, PointOBB focuses on recovering object scale and angle from point supervision for oriented detection, and Point2RBox-v2 improves point-supervised oriented detection by modeling spatial layout among instances. These methods mainly focus on oriented object detection in natural-scene images, whereas our work targets point-supervised infrared small-target detection under low-SNR conditions. Correspondingly, our technical focus is different: we study how to jointly stabilize pseudo-label evolution and rebalance the training distribution within a unified bi-level optimization framework. We will incorporate a clearer discussion of these representative works and their distinctions from our setting in the revised related work.
>
> (4) Formatting of the impact statement.
> We would like to clarify this point. According to the official ICML 2026 instructions (Call for Papers page), the main body is limited to 8 pages, while references, the Impact Statement, and appendices may appear afterward in the same PDF and do not count toward the main-body page limit. Therefore, our current placement of the Impact Statement after the 8-page main body follows the ICML 2026 formatting instructions. We will nevertheless re-check the final layout to ensure full compliance with the official template.
>
> (5) Limitation discussion.
> We thank the reviewer for pointing out the missing limitation discussion. Although Fig. 11 provides preliminary evidence that the proposed physics-inspired propagation prior can generalize across heterogeneous sparse-target modalities such as visible-band remote sensing, SAR, and sonar, we have not yet systematically evaluated the framework on a broader range of point-supervised tasks, such as point-supervised oriented object detection. We will add this point explicitly in the revised paper as a limitation and an important future direction.

---

> > ### Author Rebuttal · Reviewer_YyzY · 2026-04-03
> >
> > My concerns have been adequately addressed.

---

### Official Review · Reviewer_2sLj · 2026-03-10

**Soundness:** 3
**Presentation:** 4
**Significance:** 3
**Originality:** 3
**Overall Recommendation:** 5
**Confidence:** 5

**Summary:**

This paper discusses a bi-level optimization method to address two issues in point-supervised ISTD: the imbalanced distribution of ISTD datasets and the high cost of manual annotation. Specifically, the paper uses a meta-classifier to mitigate sample imbalance and a physics-induced diffusion process to reduce the annotation cost under point supervision. Experiments show the high efficiency of the proposed method across seven networks. Broad ablation studies indicate that this framework is stable and time-efficient, and the result is impressive: using only 30% of the dataset can achieve the best outcome, which helps alleviate the lack of ISTD datasets

**Compliance With Llm Reviewing Policy:**

Affirmed.

**Key Questions For Authors:**

1.	How do you set the warm-up epochs and the meta-classifier learning rate?
2.	Please explain more details of the proposed mechanisms and annotations refer to the weaknesses.
3.	Please explain your work’s generalizability to the general machine learning field.

**Limitations:**

This framework is efficient, but how to port this work to other machine learning fields should be discussed further. I think the bi-level method under weakly supervised settings is not only applicable to ISTD tasks.

**Strengths And Weaknesses:**

Strengths:
1.	The paper identifies key problems in ISTD: the distribution of ISTD datasets is imbalanced, and manual annotation is expensive. The motivation is good and fits the ISTD field precisely. Additionally, the framework is explained in detail in Alg. 1; the diffusion process is illustrated in Fig. 9, which is accessible.
2.	Broad experiments show high efficiency (Tables 1, 2, 4, 5, 6 and Figs. 3, 6, 8) over multiple open ISTD datasets (SIRST3, SIRST-v1, NUDT-SIRST, IRSTD-1k), improving IoU while reducing Fa. The ablation studies are compelling. Fig. 6 shows that 30% (even only 20%) of the samples can achieve SOTA results. Figs. 4 and 5 show the diffusion-based annotation method achieves both higher IoU and lower time cost, compared with COM and MCLC. Sensitivity analyses are also presented in the appendix.
3.	The discussions are wide enough. Figs. 5 and 12 show the distribution of ISTD samples, explaining the imbalance and the efficiency of the proposed method. The paper compares SAM and SAM2, which are widely used nowadays. Also, Fig. 11 in the appendix discusses more modalities (Remote Sensing, SAR, Sonar), verifying the portability of the proposed method.
4.	The paper is well formatted and readily comprehensible. Equations are clear and figures are presented neatly.
Weaknesses:
1.	The proposed method introduces many parameters. Although there is sensitivity analysis in Fig. 10, some parameters, such as warm-up epochs (set to 80 in the paper) and the meta-classifier learning rate (1 \times 10^{-3} in the paper), need to be clarified. I think these parameters are important and the paper should explain why they are set to these values.
2.	This method is based on point supervision, but how to annotate the point is unclear. “Centroid” and “Coarse” in Tables 4 and 5 are mentioned, and it needs to be clarified which setting Tables 1 and 2 use. LESPS and PAL use a Gaussian distribution to generate coarse points randomly, and I want to know whether you used the same method to imitate manual annotation.
3.	Please explain “Physics-Induced” during annotation generation. The paper uses a PDE equation to reflect a real-world meaning, but does this method really reflect the physical property?
4.	Adding a more detailed annotation generation process and the meta-classifier design to Fig. 2 would be better.
5.	Minor writing issues. Some expressions are informal, such as “definitively demonstrate” and “highly efficient”. “Direct Joint Learning” in Table 3 should be in lowercase.

---

> ### Author Rebuttal · Authors · 2026-03-30
>
> We thank the reviewer for the positive assessment of our motivation, empirical validation, and presentation quality. We are especially encouraged that the reviewer recognizes the practical importance of point-supervised ISTD. Below we respond to each concern in detail.
>
> (1)	Hyperparameters, warm-up design, and sensitivity
>
> We agree that our framework introduces several hyperparameters through the bi-level formulation. That said, these parameters are not overly sensitive in practice. Mild perturbations of the main settings do not cause a severe performance drop, so the tuning overhead is limited. In implementation, the detector is trained for 400 epochs with AdamW and learning rate $1\times 10^{-3}$; the outer-level update starts at epoch 80; the pseudo-label branch is updated every 20 epochs with learning rate $1\times 10^{-2}$; and the meta-classifier is a lightweight two-layer MLP optimized with learning rate $1\times 10^{-3}$.
>
> The rationale for the 80-epoch warm-up is to obtain a reasonably stable detector before activating the two outer branches. If bi-level updates are enabled too early, both the sample-weight branch $\alpha$ and the label-refinement branch $\theta$ may absorb excessive noise from unstable early predictions. This is especially important because our aggregated update injects the outer signal directly into detector optimization:
> $$
> \omega_{n+1}^{t}=\omega_n^{t}-\Big(\nabla_{\omega}L_{\mathrm{tr}}(\omega_n^{t},\alpha^t,\theta^t)+\beta\nabla_{\omega}L_{\mathrm{val}}(\omega_n^{t},\alpha^t,\theta^t)\Big).
> $$
> As for the meta-classifier learning rate, we found that this two-layer MLP converges quickly and behaves stably, so it does not require delicate tuning.
>
> (2)	Point annotation protocol
>
> Thank you for pointing this out. For Tables 1 and 2, we use centroid points for training. For the coarse setting in Tables 4 and 5, we follow the same configuration as LESPS and PAL. More generally, both the training and validation sets in our framework are supervised only by single-point annotations rather than dense masks.
>
> Conceptually, Centroid uses the geometric center point of the target region, whereas Coarse is intended to mimic less precise manual clicking. This distinction matters because pseudo-label generation starts from the source point $p_i$, with initialization
> $$
> u(a,b,0)=\delta(a-p_i,b-p_i),
> $$
> so the initial point quality directly affects the evolution of $\hat y_i(\theta)$.
>
> (3)	Interpretation of “Physics-Induced”
>
> Thanks for pointing this out. We agree that “Physics-Induced” may overstate the physical meaning of this module. What we really mean is that it uses a **physics-inspired** propagation mechanism as a prior for pseudo-label generation. Starting from the annotated point, the learnable anisotropic diffusion is designed to spread the signal locally in homogeneous regions while being conservative near strong boundaries and complex background structures. So the main purpose here is to make pseudo-label expansion more local, structured, and robust under point supervision, rather than to claim an exact physical model of infrared target formation.
>
> (4)	Fig. 2 and method details
>
> We appreciate this suggestion and agree that it is a fair point. The essential ingredients are already provided across the main text, Algorithm 1, and the appendix, but we agree that Fig. 2 could communicate them more directly.
> To clarify the role of each component here: the meta-classifier $g_{\psi}(\cdot)$ predicts sample-wise weights for adaptive rebalancing, while the differentiable diffusion surrogate $\mathcal{M}(\cdot;\theta)$ refines pseudo-labels using detector feedback. The update of $\theta$ is driven by
> $$
> \theta^{t+1}=\theta^t-\nabla_{\theta}\Big(L_{\mathrm{val}}(\mathcal{M}(y_i;\theta),\hat y_i)+
> L_{\mathrm{val}}(\mathcal{M}(y_i;\theta),y_i)\Big),
> $$
> where the first term aligns the surrogate with the offline pseudo-labels and the second term regularizes it by the current detector response.
>
> (5)	Writing precision
>
> We appreciate the careful reading and the writing suggestions. We agree that some expressions may sound stronger than necessary, and that a few terms can be made more precise and more formal.
>
> (6)	Generalizability beyond ISTD
>
> We also agree that the framework is not conceptually limited to ISTD. At a higher level, our method combines sparse point supervision, structured pseudo-label generation, and bi-level sample reweighting under imbalanced data. These ingredients are potentially useful for other weakly supervised dense prediction problems as well, especially when annotations are sparse and supervision quality evolves during training. In this sense, we view ISTD as the main application in this paper, but not the only possible one.

---

> > ### Author Rebuttal · Reviewer_2sLj · 2026-04-01
> >
> > My concerns have been largely resolved. Taking into account the feedback from the other reviewers and the author's responses, I will keep my score unchanged.

---

### Official Review · Reviewer_GEGB · 2026-03-13

**Soundness:** 3
**Presentation:** 3
**Significance:** 2
**Originality:** 3
**Overall Recommendation:** 5
**Confidence:** 4

**Summary:**

This paper studies infrared small target detection with point supervision. It proposes a diffusion-based pseudo-label generator, and a bi-level dual-update framework that jointly adjusts sample weights and pseudo labels during training.

**Compliance With Llm Reviewing Policy:**

Affirmed.

**Final Justification:**

The rebuttal addresses my main concerns so I raise my score from Weak Accept to Accept.

**Key Questions For Authors:**

1. A more explicit description of how the complexity map is converted into D_θ is required.
2. How does the local patch selected for pseudo-label generation?
3. Would it be possible to visualize the offline pseudo-labels more directly, and further show how they change after the joint update process?

**Limitations:**

The paper has not discussed the limitations.

In my opinion, a potential limitation is that the outer-level optimization is still supervised by physics-induced pseudo-labels on the validation set, so the refinement signal may remain biased by pseudo-label quality.

**Strengths And Weaknesses:**

Strengths:

1. The paper addresses infrared small target detection with point annotations, which is practical and annotation-efficient.
2. The method combines pseudo-label generation and sample reweighting in one training framework, which makes the overall design more coherent.
3. The work is overall solid, and the experimental section is comprehensive.

Weaknesses:

1. The relation between the offline annotator and the differentiable surrogate is not explained clearly enough, especially regarding how the learnable diffusion parameter θ is actually realized and updated in practice.
2. The optimization description in sample reweighting is a bit confusing, the main text writes the update as if α is directly optimized, while the appendix indicates that α is produced by a meta-classifier with parameters ψ.

---

> ### Author Rebuttal · Authors · 2026-03-30
>
> We sincerely thank for your constructive and encouraging comments. The concerns and questions raised are addressed in detail below.
>
> （1）	Diffusion parameterization and its practical realization:
>
> We will clarify these points more explicitly in the revision, and move the implementation details to the supplement for conciseness.
>
> (i) Explicit conversion from the complexity map to $D_\theta$.
> In the offline annotator, we first construct a local complexity map on the cropped patch around each annotated point:
> $$
> \kappa(a,b)=\alpha \|\nabla I(a,b)\|+(1-\alpha)H(a,b).
> $$
> The spatially varying diffusivity is then defined by the monotone decreasing mapping
> $$
> D_\theta(a,b)=D_0\exp\big(-\beta \kappa(a,b)\big).
> $$
> Here, $D_0$ controls the overall diffusion scale, while $\beta$ controls the suppression strength in complex regions. As a result, diffusion is stronger in homogeneous regions and weaker near cluttered structures.
>
> (ii) Relation between the offline annotator and the differentiable surrogate, and practical realization of $\theta$.
> The offline annotator and the differentiable surrogate share the same parameterization $\theta$ and the same complexity-aware diffusion principle, but use different solvers. In practice, $\theta$ is mapped to the four coefficients $(D_0,\beta,\rho,\alpha)$ through a bounded reparameterization
> $$
> (D_0,\beta,\rho,\alpha)=\Phi(\theta),
> $$
> which keeps the coefficients stable during optimization.
>
> The offline annotator constructs $D_\theta(a,b)$, performs particle-based diffusion on the cropped patch, and produces
> $$
> \hat y_i(\theta)=\rho\,u^{(K)}(a,b)+(1-\rho)\,C_{p_i}(a,b),
> $$
> which is then thresholded into $\hat y_i^{\mathrm{phys}}$. The differentiable surrogate $M(y;\theta)$ follows the same principle on the detector prediction $P=\sigma(y)$: it constructs a prediction-based complexity map, derives $D_\theta(P)$ using the same monotone mapping, performs a few diffusion steps,
> $$
> U^{(0)}=P,\qquad U^{(k+1)}=U^{(k)}+D_\theta(P)\odot \Delta U^{(k)},
> $$
> and outputs $M(y;\theta)$ after fusion with a source prior. $\theta$ is updated by backpropagating the validation objective through $M(y;\theta)$, and the updated coefficients are then used to regenerate the PDE pseudo-labels. We will add this  pipeline to the supplement.
>
> (2) Pseudo-label refinement mechanism and its limitations:
> We understand and agree with the reviewer’s concern. Our intention is not to claim that the proposed refinement fully removes the bias of the initial pseudo-labels. It is designed to reduce this bias.
> Specifically, as shown in Sec. 3.3, $\theta$ is optimized not only to align the surrogate output with the offline pseudo-labels $\hat{y}_i$, but also to maintain consistency with the current detector prediction $\mathcal{N}(x_i; {\omega})$, which makes the refinement more adaptive and alleviates overfitting to potentially biased pseudo-labels. Moreover, the refinement is guided by an additional validation split and coupled with sample reweighting, which provides an extra correction signal instead of repeatedly fitting the same labels.
> We will clarify this point explicitly in the revision and add it as a limitation of the current framework.
>
>
> (3) Sample reweighting parameterization and practical optimization:
> Thanks for pointing out this weakness. In the main text, $\alpha$ was written as the sample-weight variable for brevity. Instead, each sample weight is produced by a meta-classifier parameterized by $\psi$:
> $
> \alpha_i = \big(\sigma(g_\psi(h_i))\big),
> $. We will revise the main manuscript to explicitly state this parameterization and avoid the misleading impression.
>
> (4) Local patch selection for pseudo-label generation:
>
> We thank the reviewer for pointing out this implementation detail. We crop a local patch centered at the centroid or coarse point. The patch size is determined by the annotated target scale. Specifically, the annotator selects one of three target-size categories, namely 3$\times$ 3, 9$\times$ 9, and 15$\times$ 15 pixels, which correspond to cropped patches of 9$\times$ 9, 22$\times$ 22, and 41$\times$ 41 pixels, respectively. This patch selection policy follows MCLC (ICCV 2023). Such a design keeps pseudo-label generation local, limits the diffusion process to the neighborhood of the annotated target. We will add this implementation detail to the supplement for clarity.
>
> （5）Visualization of pseudo-label generation and online stability:
> Thank you for this helpful suggestion. We have provided more direct visualizations of the offline pseudo-labels and the stability of online updates in an anonymous link(https://anonymous.4open.science/r/anonymous-1405/). In addition, we have included an online performance table.
>
> |Epoch|w/o|Ours|
> |--:|--:|--:|
> |40|53.42|53.33|
> |80|61.86|61.94|
> |120|65.47|67.03|
> |160|68.30|70.25|
> |200|69.20|71.21|
> |240|70.81|72.07|
> |280|71.11|72.57|
> |320|71.38|72.93|
> |360|71.46|73.10|
> |400|71.52|73.28|
>
> Training on SIRST3 with ALCLNet. Configuration is same as main experiment.

---

> > ### Author Rebuttal · Reviewer_GEGB · 2026-04-03
> >
> > The rebuttal addresses my main concerns well. I encourage the authors to make the implementation details and the optimization flow more explicit in the revised version. I will raise my score from Weak Accept to Accept.

---

### Decision · Program_Chairs · 2026-04-30

**Decision:**

Accept (spotlight)

**Comment:**

The paper addresses infrared small target detection with point annotations. The paper mainly handles with the two challenges of unstable pseudo-label evolution in low-SNR infrared imagery and severe sample-distribution imbalance, and proposes a bi-level dual-update optimization framework and a physics-induced diffusion model expanding point annotations into pseudo-masks.
All the reviewers found that the paper has merits for the community, and the rebuttal satisfactory addressed their concerns. The AC agrees with the reviewers’ consensus to accept this paper.